# A model of COVID-19 pandemic with vaccines and mutant viruses

Young Rock Kim[1]☯, Yong-Jae Choi[2]☯, Youngho Min[1]☯*

**1** Major in Mathematics Education, Graduate School of Education, Hankuk University of Foreign Studies, Seoul, Republic of Korea, **2** Economics Division, Hankuk University of Foreign Studies, Seoul, Republic of Korea

☯ These authors contributed equally to this work.
* pumpzxc@gmail.com

**Data Availability Statement:** The daily infection data and the efficacy of the booster shot were obtained from the Korea Disease Control and Prevention Agency (KDCA) press release (Available from https://kdca.go.kr/board/board.es?mid=a20501010000&bid=0015). Vaccine data were

## Abstract

This paper proposes a compartment model (*SVEIHRM* model) based on a system of ordinary differential equations to simulate the pandemic of severe acute respiratory syndrome coronavirus 2 (SARS-CoV-2).Emergence of mutant viruses gave rise to multiple peaks in the number of confirmed cases. Vaccine developers and WHO suggest individuals to receive multiple vaccinations (the primary and the secondary vaccinations and booster shots) to mitigate transmission of COVID-19. Taking this into account, we include compartments for multiple vaccinations and mutant viruses of COVID-19 in the model. In particular, our model considers breakthrough infection according to the antibody formation rate following multiple vaccinations. We obtain the effective reproduction numbers of the original virus, the Delta, and the Omicron variants by fitting this model to data in Korea. Additionally, we provide various simulations adjusting the daily vaccination rate and the timing of vaccination to investigate the effects of these two vaccine-related measures on the number of infected individuals. We also show that starting vaccinations early is the key to reduce the number of infected individuals. Delaying the start date requires increasing substantially the rate of vaccination to achieve similar target results. In the sensitivity analysis on the vaccination rate of Korean data, it is shown that a 10% increase (decrease) in vaccination rates can reduce (increase) the number of confirmed cases by 35.22% (82.82%), respectively.

## Introduction

Severe acute respiratory syndrome coronavirus 2 (SARS-CoV-2), also known as COVID-19, emerged in late 2019 [1] and continues to threaten the world [2]. The virus has reduced social activities such as traveling and interactions between individuals [3] and negatively impacted education [4] and the economy [5]. By the end of April 2022, worldwide, there had been more than five hundred million confirmed infections and six million deaths [2]. The World Health Organization (WHO) declared a global public health emergency against COVID-19 on January 3, 2020, and a pandemic on March 12, 2020 [6].

Researchers have been concerned about the risk of COVID-19 from the early stage of the outbreak [7] and have proposed non-pharmaceutical interventions such as wearing a mask, social distancing, and quarantine policies [8–11]. Korea is considered one of the countries that

obtained from the COVID-19 dashboard of the KDCA (Available from http://ncov.mohw.go.kr/en/). All data are publicly available.

**Funding:** This work was supported by the Hankuk University of Foreign Studies Research Fund. Young Rock Kim and Youngho Min were supported by the National Research Foundation of Korea (NRF) grant funded by the Korea government (MSIT) (No. 2021 R1A2C1011467). The funders had no role in study design, data collection and analysis, decision to publish, or preparation of the manuscript.

**Competing interests:** The authors have declared that no competing interests exist.

have followed non-pharmaceutical interventions well [12, 13]. On February 23, 2020, the Korea Disease Control and Prevention Agency (KDCA) quickly confirmed the increasing number of cases and issued a high-level concern (red alert) for COVID-19 [14]. From the early stage of COVID-19, the Korean government implemented policies such as screening individuals contacted with confirmed cases, immediate isolation of all confirmed cases, school closures, and a variety of social-distancing measures. The efficacy of these measures has been outstanding and has prevented a rapid increase in the number of COVID-19 cases [13, 15]. Thereafter, COVID-19 vaccines were developed and populations were inoculated, but Korea faced a new phase when variants of COVID-19 emerged.

Mathematical models are useful for predicting how the disease spread and for designing intervention policies [16–19]. The susceptible-infected-removed (*SIR*) model focuses on disease compartments that was developed in 1927 by Kermack and McKendrick and has been used as a basic mathematical model [20]. The *SIR* model has been expanded to consider an incubation period, reinfection, and quarantining, see [21–24] for more details.

Two types of extensions are of particular relevance to our study. The first are multi-strain models that consider pathogen mutations. Gubar et al. used a model with two strains to study the propagation processes of different strains of influenza viruses [25]. They assumed that viruses with higher virulence are more likely to succeed in spreading when an infected individual interacts with one who is susceptible. They modeled two strains with different transmission rates and recovery periods. Conversely, Gordo et al.'s model considered the relationship between two strains, and they set up a model in which an individual infected with an original virus is infected again with a mutant virus [26]. A susceptible-exposed-infected-recovery (*SEIR*) model involving two strains was proposed by Khayar and Allali [27]. It is assumed that individuals infected with one strain develop immunity against mutants of the same strain. Arruda et al. also proposed a *SEIR*-based multi-strain model for an arbitrary number of strains considering reinfection with the same strain as well as reinfection with different strains [28]. Lazebnik et al. generalized the multi-strain model by considering all possible infection sequences for an arbitrary number of strains [29]. They provided a variety of simulations to estimate the epidemiological characteristics of an epidemic such as mortality rate, maximum infection rate, and an average basic reproduction number.

The second type of extension considers vaccination. Fudolig et al. considered a multi-strain epidemiological model with selective immunity by vaccination [30]. The authors focused on examining the effects of introducing a new strain on the population when the existing strain has reached equilibrium. Lazebnik et al. provided an advanced multi-mutation model that considers intervention policies including vaccination and lockdown [31].

We worked on a COVID-19 pandemic model with multiple mutant viruses and studied the effect of vaccination on disease propagation in Korea. We developed a mathematical model based on the *SEIR* type of compartment model into which we added various compartments related to multiple vaccinations and mutant viruses of COVID-19. The difference from other studies is that the multiple vaccination compartments were designed to consider the antibody formation rates of the original virus and the mutant virus according to the number of vaccinations. Additionally, to consider breakthrough infections, we assumed that vaccinated individuals can also become infected with the original or mutated strain if they do not develop antibodies even after being vaccinated. We calculated the effective reproduction numbers of the original virus and the Delta variant. Through this, we found the critical level of effective reproduction number over which the mutant virus becomes prevalent using Korean data. The strength of our model is that we control the proportion of susceptible individuals who transfer to vaccination compartments daily, allowing simulations with the number of daily vaccinations and initial inoculation date. As a specific example, we showed the efficacy of the vaccine

by providing a sensitivity analysis that changed the vaccination rate through model fitting with actual data obtained between March 12, 2020 and December 31, 2021 in Korea. We hope that the results of these simulations help make rational decision-making by those who are reluctant to vaccination.

## Methods

Diverse mathematical models exist for infectious diseases [32]. The compartment model is one of the representative mathematical modeling techniques [11]. In this section, we introduce a mathematical model that shows the effect of vaccinations on the transmission of COVID-19 and its variants. We added compartments for primary and secondary vaccinations and booster shots to the multi-strain model based on *SEIHR*, considering the vaccine's rate of antibody formation. We first introduced two types of transitions to be used in the model.

### Two types of transitions

There are two types of transitions in Fig 1 to understand epidemic models [33]. The arrows in Fig 1 indicate transitions between compartments. One is nonspontaneous transition and the other is spontaneous transition. A nonspontaneous transition occurs when a compartment $X$

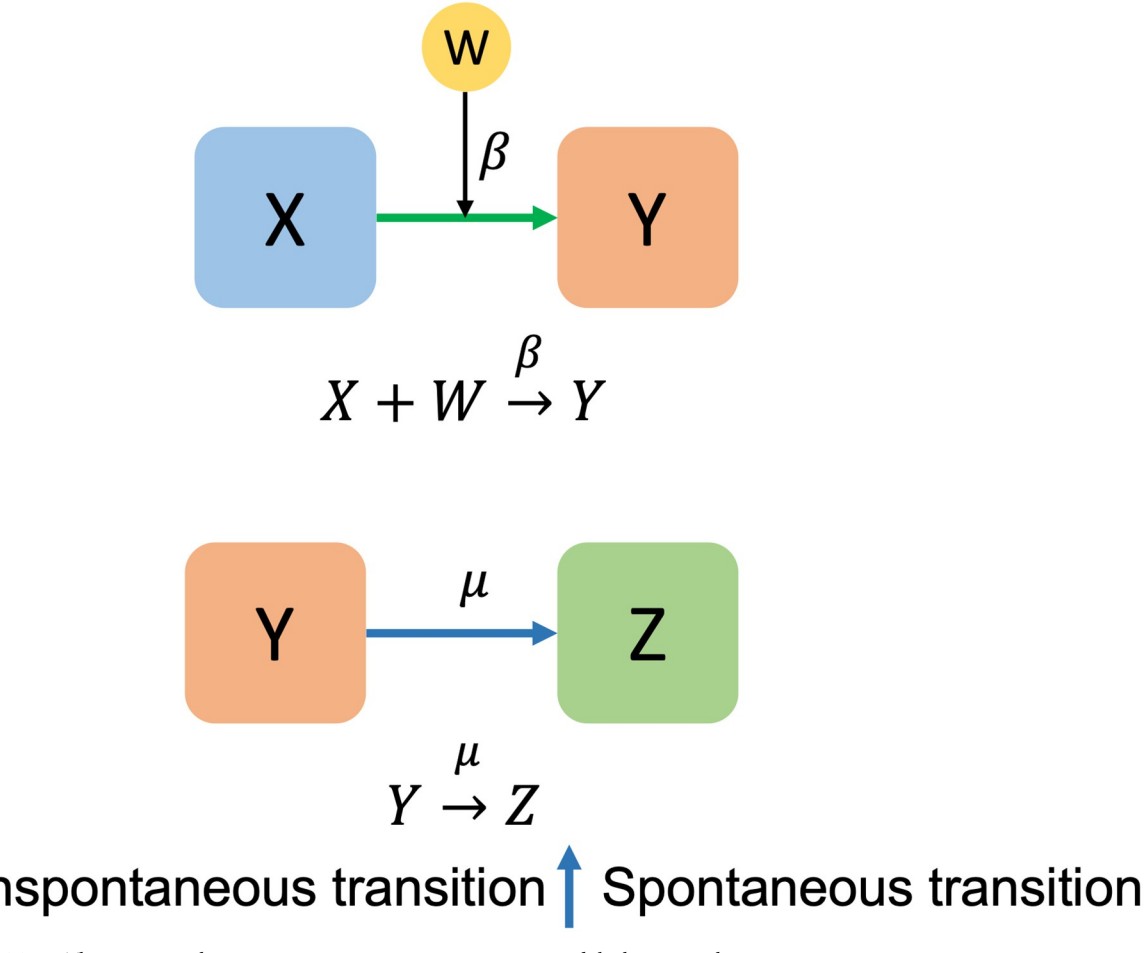

**Fig 1. Two transitions.** The upper panel represents a non-spontaneous transition and the lower panel represents a spontaneous transition.

is converted into a compartment $Y$ at a rate of $\beta$ under the influence of a compartment $W$. Conversely, spontaneous transition means that a compartment $Y$ is converted into a compartment $Z$ at the rate of $\mu$ without the influence of other compartments. In epidemiological models, shifts between compartments are usually represented by the above two types of transitions in epidemiological models using differential equations.

### *SVEIHRM* model

In this subsection, we introduce a model that adds several compartments, such as multi-vaccination compartments, using the above two transitions to the multi-strain *SEIR* model. Before setting up the model, we present the assumptions used in the model.

1. Because the COVID-19 pandemic has manifested for a long period, our model considers the natural mortality and birth rates of the population in the course of the pandemic.

2. An individual who has been vaccinated but has not developed antibodies can be infected. When a vaccine does not provide complete immunity to the virus and the vaccinated individual becomes infected with the disease, it is called "breakthrough infection" [34].

3. Individuals infected with one strain develop immunity against other mutations of the same strain. That is, we rule out reinfection, which means an individual who has recovered from the original virus becomes immune to the mutant virus and vice versa. This assumption was adopted in [27]. Reinfection with the same strain is also excluded.

4. Infection by a mutant virus occurs through contact with an individual infected with a mutant virus and infection by the original virus occurs through contact with an individual infected with the original virus. More specifically, a susceptible individual who comes into contact with an individual infected with the original virus may or may not become infected with the original virus, but they do not become an individual infected with the mutant virus.

5. Hospitalized individuals include those who are hospitalized as well as those who are self-isolating. Individuals in this group do not develop secondary infections.

6. The vaccine's efficacy against the mutant is slightly inferior than that against the original virus [35–38].

To set up the *SVEIHRM* model, we added vaccinated compartments, mutant virus compartments, and hospitalized compartments to the *SEIR* model. First, we consider multiple vaccinations by adding $V_1$, $V_2$, and $V_3$ compartments to represent the groups with primary, secondary vaccination, and booster shots, respectively. We also added $E_m$ and $I_m$ compartments to the groups exposed to the mutant virus and infected with the mutant virus, respectively. The compartment for the hospitalized group and the compartment for the recovered group are divided into three categories according to the symptoms of the patient: mild, moderate, or severe. Fig 2 is a diagram of the *SVEIHRM* model, and all the nodes are summarized below.

- *S* (Susceptible group): The compartment for susceptible individuals. When a susceptible individual encounters an infected individual, the susceptible individual can be converted into the exposed group.

- $V_1$ (A group with primary vaccination): The compartment for individuals with primary vaccination. Vaccines for COVID-19 include products from AstraZeneca, Pfizer, and Moderna. The effectiveness of primary vaccines is taken to be the average of the preventive effects of the three types of vaccines [39, 40].

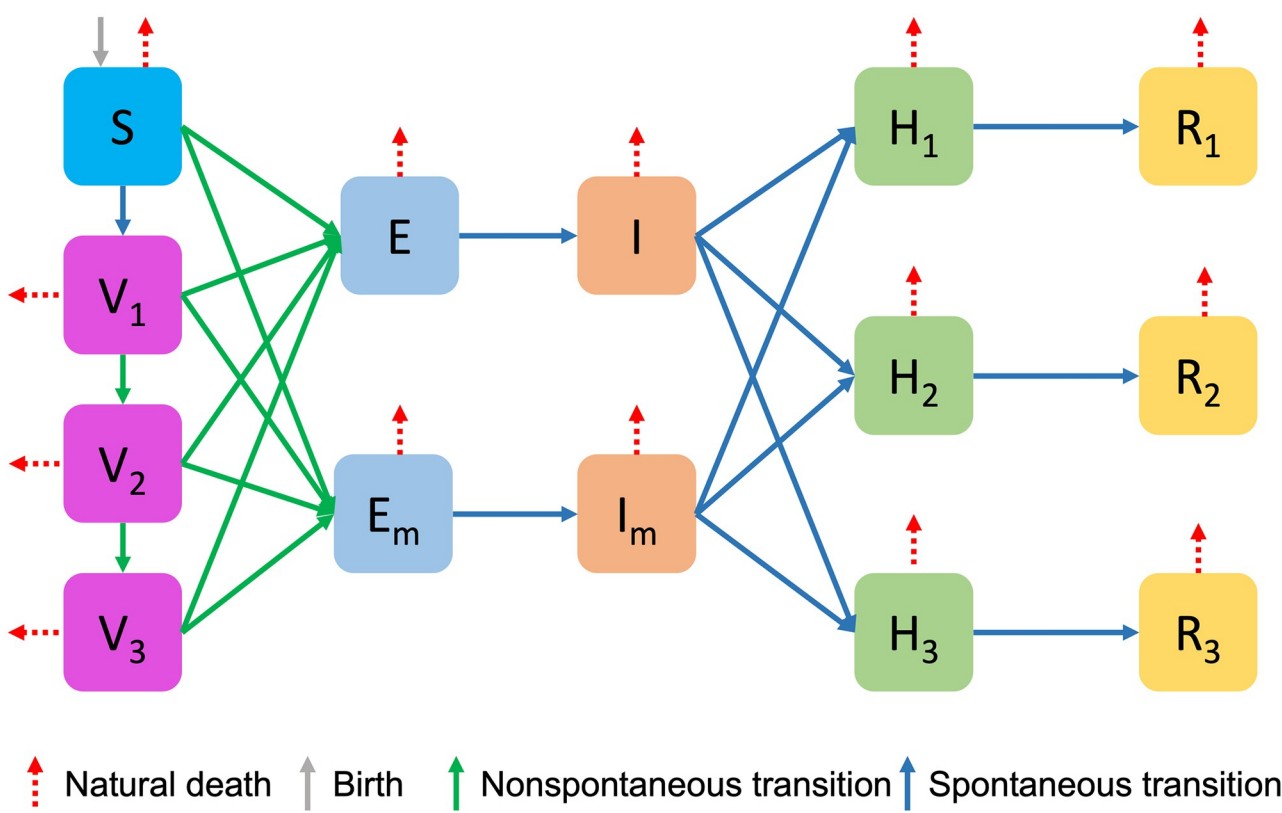

**Fig 2. A Diagram of a *SVEIHRM* Model.**

- $V_2$ (A group with secondary vaccination): The compartment for individuals with secondary vaccination. The effectiveness of secondary vaccines is taken to be the average of the preventive effects of the three types of vaccines [39, 40]. (Janssen inoculations are classified as secondary vaccination.)

- $V_3$ (A group with booster shots): The compartment for individuals with booster shots. The effectiveness of booster shots is taken to be the average of the preventive effects of the three types of vaccines [41].

- $E$ (A group exposed to the original virus): The compartment for individuals exposed to the original virus. Individuals in this group go through an incubation period and move to the group infected with the original virus.

- $E_m$ (A group exposed to the mutant virus): The compartment for individuals exposed to the mutant virus. This group moves through the incubation period to the group infected with the mutant virus.

- $I$ (A group of infected with the original virus): The compartment for individuals infected with the original virus.

- $I_m$ (A group of infected with the mutant virus): The compartment for individuals infected with the mutant virus.

- $H_k$ (A hospitalized group) ($k = 1, 2, 3$): The compartments for inpatients in the hospital. In the order of ($k = 1, 2, 3$), each compartment is a group of mild, moderate, or severe patients.

**Table 1. Entering and exiting transitions of each node.**

| Node | Entering | Entering Rate | Transition's type | Exiting | Exiting rate | Transition's type |
|---|---|---|---|---|---|---|
| $S$ | - | - | - | $S \to V_1$ | $\alpha_1(t)$ | Spontaneous |
| | | | | $S \to E$ | $\beta$ | Nonspontaneous |
| | | | | $S \to E_m$ | $\beta^m$ | Nonspontaneous |
| $E$ | $S \to E$ | $\beta$ | Nonspontaneous | $E \to I$ | $\sigma$ | Spontaneous |
| | $V_1 \to E$ | $\delta_1 \times \beta$ | Nonspontaneous | | | |
| | $V_2 \to E$ | $\delta_2 \times \beta$ | Nonspontaneous | | | |
| | $V_3 \to E$ | $\delta_3 \times \beta$ | Nonspontaneous | | | |
| $E_m$ | $S \to E_m$ | $\beta^m$ | Nonspontaneous | $E_m \to I_m$ | $\sigma$ | Spontaneous |
| | $V_1 \to E_m$ | $\delta_1^m \times \beta^m$ | Nonspontaneous | | | |
| | $V_2 \to E_m$ | $\delta_2^m \times \beta^m$ | Nonspontaneous | | | |
| | $V_3 \to E_m$ | $\delta_3^m \times \beta^m$ | Nonspontaneous | | | |
| $I$ | $E \to I$ | $\sigma$ | Spontaneous | $I \to H_k$ | $\eta_k \times \mu$ | Spontaneous |
| $I_m$ | $E_m \to I_m$ | $\sigma$ | Spontaneous | $I_m \to H_k$ | $\eta_k^m \times \mu$ | Spontaneous |
| $H_k$ | $I \to H_k$ | $\eta_k \times \mu$ | Spontaneous | $H_k \to R_m$ | $\gamma_k$ | Spontaneous |
| | $I_m \to H_k$ | $\eta_k^m \times \mu$ | Spontaneous | | | |
| $R_k$ | $H_k \to R_k$ | $\gamma_k$ | Spontaneous | - | - | - |
| $V_1$ | $S \to V_1$ | $\alpha_1(t)$ | Spontaneous | $V_1 \to V_2$ | $\alpha_2(t)$ | Spontaneous |
| | | | | $V_1 \to E$ | $\delta_1 \times \beta$ | Nonspontaneous |
| | | | | $V_1 \to E_m$ | $\delta_1^m \times \beta^m$ | Nonspontaneous |
| $V_2$ | $V_1 \to V_2$ | $\alpha_2(t)$ | Spontaneous | $V_2 \to V_3$ | $\alpha_3(t)$ | Spontaneous |
| | | | | $V_2 \to E$ | $\delta_2 \times \beta$ | Nonspontaneous |
| | | | | $V_2 \to E_m$ | $\delta_2^m \times \beta^m$ | Nonspontaneous |
| $V_3$ | $V_2 \to V_3$ | $\alpha_3(t)$ | Spontaneous | $V_3 \to E$ | $\delta_3 \times \beta$ | Nonspontaneous |
| | | | | $V_3 \to E_m$ | $\delta_3^m \times \beta^m$ | Nonspontaneous |

Table 1 shows the transitions entering and exiting each node and each transition's type.

As stated above, hospitalized individuals are quarantined and do not transmit the virus to the susceptible.

- $R_k$ (A removed group) ($k = 1, 2, 3$): The compartment for removed (and immune) or deceased individuals. This compartment may also be called named "recovered" or "resistant". Since the severity of symptoms varies, the recovery period is differs according to $k$.

Table 1 shows the transitions entering and exiting each node. Also, the meanings of parameters for incoming and outgoing rates used in Table 1 are described in Table 2. As shown in Table 1, individuals belonging to the susceptible group $S$ move to the group exposed to the original virus or the mutant virus and the group vaccinated. When the transmission rate of the original virus is $\beta$, the mutant virus is set to $\beta^m = \tau \times \beta$. This allows $\tau$ to determine how fast the mutant virus is spreading proliferating compared to the original virus.

There are four cases where a susceptible individual moves to the exposed group. One is a case in which the susceptible individuals are exposed to the virus, and the others are cases in which the individuals who received primary and secondary vaccinations and booster shots are exposed to the virus. When the vaccinated group is exposed, those who do not form antibodies ($\delta_k V_k$) can become infected at a rate of $\beta$ ($k = 1, 2, 3$). Therefore, in our model, the number of individuals susceptible to the original virus can be expressed as $S$, $\delta_1 V_1$, $\delta_2 V_2$ and $\delta_3 V_3$. Similarly, the number of individuals susceptible to the mutant virus can be expressed as $S$, $\delta_1^m V_1$, $\delta_2^m V_2$ and $\delta_3^m V_3$, where the transmission rate is $\beta^m$ instead of $\beta$. Infectious individuals are

**Table 2. Descriptions of parameters.**

| Parameters | Descriptions |
|---|---|
| $\Lambda$ | The Number of births |
| $v$ | Natural death rate |
| $\beta$ | Transmission rate of original virus |
| $\beta^m = \tau\beta$ | Infection transmission rate of mutant virus |
| $1/\sigma$ | Period from diagnosis of infection to isolation |
| $1/\mu$ | The period that it takes for an infected individual to be hospitalized |
| $1/\gamma_1, 1/\gamma_2, 1/\gamma_3$ | Days for recovery of mild, moderate, or severe patients |
| $\alpha_1(t), \alpha_2(t), \alpha_3(t)$ | Primary, secondary and booster shots vaccination rates |
| $\delta_1, \delta_2, \delta_3$ | Probability that the primary, secondary vaccine and booster shot recipients do not form antibodies to original virus after 28 days of inoculation |
| $\delta_1^m, \delta_2^m, \delta_3^m$ | Probability that the primary, secondary vaccine and booster shot recipients do not form antibodies to mutant viruses after 28 days of inoculation |
| $\eta_1, \eta_2, \eta_3$ | Proportion of original virus infected patients hospitalized with mild, moderate, or severe symptoms |
| $\eta_1^m, \eta_2^m, \eta_3^m$ | Proportion of mutant virus infected patients hospitalized with mild, moderate, or severe symptoms |

Table 2 notes descriptions of parameters for entering and exiting rates used in Table 1.

hospitalized and are spontaneously converted. Here, the hospitalized compartments and recovered compartments that are classified into three cases can be used in various ways. For example, since the treatment cost for each symptom is different, it can be used to analyze the economic effect. We can also calculate the number of beds in hospitals according to the hospitalization period. Note that individuals with mild symptoms should self-isolate even if they are not hospitalized by model assumption 5. Using the transmission rate and transition type of each node, and taking births and natural deaths into account, we can derive the following system of differential equations:

$$\frac{d}{dt}S(t) = \Lambda - \beta\frac{S(t)I(t)}{N} - \beta^m\frac{S(t)I_m(t)}{N} - \alpha_1(t)S(t) - vS(t)$$

$$\frac{d}{dt}E(t) = \beta\frac{S(t)I(t)}{N} + \delta_1\beta\frac{V_1(t)I(t)}{N} + \delta_2\beta\frac{V_2(t)I(t)}{N} + \delta_3\beta\frac{V_3(t)I(t)}{N} - \sigma E(t) - vE(t)$$

$$\frac{d}{dt}E_m(t) = \beta^m\frac{S(t)I_m(t)}{N} + \delta_1^m\beta^m\frac{V_1(t)I_m(t)}{N} + \delta_2^m\beta^m\frac{V_2(t)I_m(t)}{N} + \delta_3^m\beta^m\frac{V_3(t)I_m(t)}{N}$$
$$\qquad - \sigma E_m(t) - vE_m(t)$$

$$\frac{d}{dt}I(t) = \sigma E(t) - (\eta_1 + \eta_2 + \eta_3)\mu I(t) - vI(t)$$

$$\frac{d}{dt}I_m(t) = \sigma E_m(t) - (\eta_1^m + \eta_2^m + \eta_3^m)\mu I_m(t) - vI_m(t) \qquad (1)$$

$$\frac{d}{dt}H_k(t) = \eta_k\mu I(t) + \eta_k^m\mu I_m(t) - \gamma_k H_k(t) - vH_k(t)$$

$$\frac{d}{dt}R_k(t) = \gamma_k H_k(t) - vR_k(t)$$

$$\frac{d}{dt}V_1(t) = \alpha_1(t)S(t) - \alpha_2(t)V_1(t) - \delta_1\beta\frac{V_1(t)I(t)}{N} - \delta_1^m\beta^m\frac{V_1(t)I_m(t)}{N} - vV_1(t)$$

$$\frac{d}{dt}V_2(t) = \alpha_2(t)V_1(t) - \alpha_3(t)V_2(t) - \delta_2\beta\frac{V_2(t)I(t)}{N} - \delta_2^m\beta^m\frac{V_2(t)I_m(t)}{N} - vV_2(t)$$

$$\frac{d}{dt}V_3(t) = \alpha_3(t)V_2(t) - \delta_3\beta\frac{V_3(t)I(t)}{N} - \delta_3^m\beta^m\frac{V_3(t)I_m(t)}{N} - vV_3(t),$$

where $k = 1, 2, 3$, $\eta_1 + \eta_2 + \eta_3 = 1$, and $\beta^m = \tau\beta$. Details of the mathematical model are provided in S1 Text. In the next subsection, we will define the disease-free equilibrium (DFE) state using the model's infectious compartments of this model. We will also derive the basic reproduction number and the effective reproduction number for the original virus and mutant viruses.

## Reproduction numbers of the mathematical model

The compartment model consists of a system of differential equations representing the transition flow between partitioned groups, such as the *SIR* model and the *SEIR* model. However, we have difficulties in classifying or notating the compartments when there are many compartments. Assuming that a collection of all compartments is $\mathcal{C}$, we define four subcollections according to the characteristics of the compartments in Table 3.

Assume that the compartment model has $n$ compartments. When the population belonging to each compartment is $X_i$ for $0 \leq i \leq n$, model $X$ can be written as

$$X = (X_1, X_2, \cdots, X_n).$$

Now we divide each compartment into two types: infectious compartments and a noninfectious compartments. The distinction between infectious and noninfectious compartments cannot be deduced from the structure of the equations alone. By epidemiological interpretation of the model, we assign compartments in $\mathcal{C}^i$ to infectious compartments and compartments in $\mathcal{C}\backslash\mathcal{C}^i$ to noninfectious compartments. We define a set of DFE states as $X_{\mathrm{DFE}} = \{X \geq 0 | \text{all elements of} \mathcal{C}^i \text{are zero}\}$. The reproduction numbers of the compartment model depends on the infectious and noninfectious compartments of a model.

**The basic reproduction number.** The basic reproduction number represents the average number of secondary infections that the first infected individual can infect in DFE state. This is a numerical indicator of the speed at which an infectious disease spreads. According to Macdonald, G., an epidemic breaks out when $R_0 > 1$ and it disappears naturally when $R_0 < 1$ [42]. Therefore, finding the $R_0$ of a disease is an important task in analyzing and simulating the epidemic model. Van den Driessche, P. and Watmough, J. presented a method to find $R_0$ of a complex compartment epidemic model, which is to find the spectral radius of the next generation matrix $G$ in the DFE state, ie, $R_0 = \rho(G)$ [43]. The method for deriving the next generation matrix $G$ is described below.

We now derive the basic reproduction number of our model. The basic reproduction number is dependent on the DFE at the beginning of the epidemic. Vaccines for COVID-19 was developed after a certain period of time after the outbreak. Also, in the early stage of an

**Table 3. Collections of compartments.**

| Collections | Description | Elements |
|---|---|---|
| $\mathcal{C}$ | A collection of all compartments | $S, E, I, E_m, I_m, H_1, H_2, H_3, R_1, R_2, R_3, V_1, V_2, V_3$ |
| $\mathcal{C}^s$ | A collection of susceptible compartments | $S, V_1, V_2, V_3$ |
| $\mathcal{C}^i$ | A collection of infectious compartments | $I, E, I_m, E_m$ |
| $\mathcal{C}^c$ | A collection of compartments that can infect original virus | $I, E$ |
| $\mathcal{C}^m$ | A collection of compartments that can infect mutant virus | $I_m, E_m$ |

Table 3 notes collections of specific compartments. $\mathcal{C}$ represents the collection of of all compartments and the remaining four collections represent subcollections of $\mathcal{C}$ according to the characteristics of the compartments.

outbreak, there are no mutant viruses of COVID-19 so we do not need to consider the compartments $V_1$, $V_2$, $V_3$, $E_m$ and $I_m$. As a result, our model at the DFE state is represented by the *SEIHR* model without regarding vaccinations and mutant viruses:

$$
\begin{aligned}
\frac{d}{dt} S(t) &= \Lambda - \beta \frac{S(t)I(t)}{N} - vS(t) \\
\frac{d}{dt} E(t) &= \beta \frac{S(t)I(t)}{N} - \sigma E(t) - vE(t) \\
\frac{d}{dt} I(t) &= \sigma E(t) - (\eta_1 + \eta_2 + \eta_3)\mu I(t) - vI(t) \\
\frac{d}{dt} H_k(t) &= \eta_k \mu I(t) + \eta_k^m \mu I_m(t) - \gamma_k H_k(t) - vH_k(t) \\
\frac{d}{dt} R_k(t) &= \gamma_k H_k(t) - vR_k(t)
\end{aligned}
\tag{2}
$$

where $k = 1, 2, 3$ and $\eta_1 + \eta_2 + \eta_3 = 1$.

Let $X$ be the model of *SEIHR*, that is $X = (S, E, I, H_1, H_2, H_3, R_1, R_2, R_3)$. We consider the $I$ and $E$ compartments of $X$ as infectious compartments. A DFE state of $X$ can be defined as $D^0 = (S, 0, 0, 0, 0, 0, 0, 0, 0)$ where $S = N$. To derive the next generation matrix $G$, we define a matrix $F$ which is the rate of new individuals into the infectious $E$ and $I$, and a matrix $V$ which is the transfer rate of individuals from the $E$ and $I$ compartments to the noninfectious $H$ and $R$ compartments. According to [43], the next generation matrix $G$ is defined as $FV^{-1}$ and we can estimate matrices $F$ and $V$ as follows:

$$
F = \begin{pmatrix} \beta & 0 \\ 0 & 0 \end{pmatrix}, \quad V = \begin{pmatrix} 0 & \sigma \\ \mu & -\sigma \end{pmatrix}.
\tag{3}
$$

Then we can obtain the next generation matrix

$$
\begin{aligned}
G = FV^{-1} &= -\frac{1}{\sigma\mu} \begin{pmatrix} \beta & 0 \\ 0 & 0 \end{pmatrix} \begin{pmatrix} -\sigma & -\sigma \\ -\mu & 0 \end{pmatrix} \\
&= \begin{pmatrix} \frac{\beta}{\mu} & \frac{\beta}{\mu} \\ 0 & 0 \end{pmatrix}.
\end{aligned}
\tag{4}
$$

Let $\rho(A)$ be a spectral radius of the matrix $A$. According to [43], the spectral radius (the largest absolute value of eigenvalues of the matrix) of the next generation matrix $G$ is the basic reproduction number, then we can obtain $R_0$ such that

$$
R_0 = \rho(G) = \frac{\beta}{\mu},
\tag{5}
$$

where $\beta$ is the transmission rate and $\mu$ is the recovered or removed rate.

**Proposition 1**. *If $D^0$ is the DFE state of the system (2), then $D^0$ is locally asymptotically stable if $R_0 = \rho(G) \leq 1$, but unstable if $R_0 > 1$.*

*Proof.* The proof of this proposition proceeds similarly to the derivation of the basic reproduction number of the *SEIR* model of the study [44]. Details of the proof are provided in S2 Text.

Proposition 1 shows the stability of the solution according to $R_0$ in the DFE state.

**The effective reproduction number.** Similar to the basic reproduction number, the number of secondary infections produced by one infected individual after the initial outbreak is called the effective reproduction number [45]. The effective reproduction number at time $t$ is denoted by $R_t$. It is used to measure the immediate infectiousness level of the disease at time $t$. The effective reproduction number can be derived in the similar way as the basic reproduction number. In previous subsection, we assumed that all populations were susceptible and there was no vaccine. Here, we assume that some individuals are already infected and a vaccine is developed to obtain the effective reproduction number at time $t$. Then the matrix $F$ which is the rate of new individuals into the infectious $E$ and $I$ is altered by the assumptions. We have to consider not only individuals infected in the $S$ compartment, but also individuals infected in the $V_1$, $V_2$ and $V_3$ compartments. Using this modified $F$ matrix, we can obtain the next generation matrix $G$ to obtain the effective reproduction number [45].

In our model, infectious compartments are divided into two categories according to the type of virus. One is infectious compartments of original virus ($E$ and $I$) and another is infectious compartments of mutant virus ($E_m$ and $I_m$). First, we calculate the effective reproduction number of original virus. We define a matrix $F_1$ which is the rate of new individuals into the infectious compartments $E$ and $I$, and a matrix $T_1$ which is the rate of transformation of individuals from the $E$ and $I$ compartments to the non-infectious compartments.

$$F_1 = \begin{pmatrix} \beta\left(\dfrac{S + \delta_1 V_1 + \delta_2 V_2 + \delta_3 V_3}{N}\right) & 0 \\ 0 & 0 \end{pmatrix}, \quad T_1 = \begin{pmatrix} 0 & \sigma \\ \mu & -\sigma \end{pmatrix}. \tag{6}$$

Using $F_1$ and $T_1$, we can obtain the spectral radius of the next generation matrix $G_1 = F_1 T_1^{-1}$. Then, the effective reproduction number of original virus is as follows

$$\rho(G_1) = R_t = \frac{\beta}{\mu} \times \frac{S + \delta_1 V_1 + \delta_2 V_2 + \delta_3 V_3}{N}. \tag{7}$$

This effective reproduction number has the following relationship with $R_0$,

$$R_t = R_0\left(\frac{S + \delta_1 V_1 + \delta_2 V_2 + \delta_3 V_3}{N}\right). \tag{8}$$

This is a form obtained by multiplying the basic reproduction number by the population that can be infected among the total population. Here, since $(S + \delta_1 V_1 + \delta_2 V_2) \leq N$, then $R_t \leq R_0$ always holds.

Similarly, we define $F_m$ and $T_m$ for mutant virus. The matrix $F_m$ is the rate of new individuals into the infectious compartments $E_m$ and $I_m$, and the matrix $T_m$ is the rate of transfer of individuals from the $E_m$ and $I_m$ compartments to the non-infectious compartments. These matrices are computed as

$$F_m = \begin{pmatrix} \beta^m\left(\dfrac{S + \delta_1^m V_1 + \delta_2^m V_2 + \delta_3^m V_3}{N}\right) & 0 \\ 0 & 0 \end{pmatrix}, \quad T_m = \begin{pmatrix} 0 & \sigma \\ \mu & -\sigma \end{pmatrix}. \tag{9}$$

Then the effective reproduction number of mutant virus is as follows

$$\rho(G_m) = R_t^m \quad = \frac{\beta^m}{\mu} \times \frac{S + \delta_1^m V_1 + \delta_2^m V_2 + \delta_3^m V_3}{N}$$

$$= R_0 \times \tau\left(\frac{S + \delta_1^m V_1 + \delta_2^m V_2 + \delta_3^m V_3}{N}\right). \tag{10}$$

The effective reproduction number of the mutant virus is a form of multiplying $R_0$ by the number of infected individuals out of the total number of population and $\tau$ value which is representing the transmission rate of the mutant virus.

## Results

This section provides several simulations using the *SVEIHRM* model. For the simulations, we used the code and library written in the MATLAB program. We examine the number of infected individuals while adjusting the parameters related to the vaccines, original, and mutant virus.

We considered the following three cases:

**Case 1:** Original virus without vaccination.

**Case 2:** Original virus with vaccination.

**Case 3:** Original virus with mutant virus and vaccinations.

The above cases represent the proliferation of COVID-19. The Case 1 shows the early stage of the COVID-19 outbreak, when vaccines and mutant viruses do not exist. In Case 2, vaccines have been developed, and we focus on analyzing the impact of vaccination on the number of infected individuals. Finally, we add a mutant virus to the Case 2 to simulate a Case 3 with vaccinations and mutant viruses.

### Case 1: Original virus without vaccination

In the *SVEIHRM* model, if the parameters for compartments related to vaccines and mutant virus ($V_1$, $V_2$, $V_3$, $E_m$, and $I_m$) are set to 0, the situation in which there is no vaccine or mutant virus can be simulated. That is, parameters $\delta_k$ and $\alpha_k$ ($k = 1, 2, 3$) related to the vaccine and parameters $\beta^m$, $\delta_k^m$, and $\eta_k^m$ (k = 1,2,3) related to the mutant virus are set to 0. Then our model is simply expressed as the *SEIHR* model as in Eq (2). The basic reproduction number of the *SEIHR* model is $R_0 = \beta/\mu$ (5). We assumed the total population to be $N = 50, 000, 000$, $\sigma = 1/4.1$, and $\mu = 1/4$, which will be commonly used in this subsection [46, 47]. Additionally, the proportions of infected individuals hospitalized for mild, moderate, or severe symptoms were set to 80%, 18%, and 2%, respectively. The values of commonly used parameters were shown in Table 4. Four results were obtained by changing $R_0$ from 1.5 to 3 by 0.5. Fig 3 shows the simulation result.

Following the simulation, $R_0$ was found to increase, and the increase in the number of infections was faster with the maximum number of daily infections being larger. The total number of infected individuals is represented by the area under the graph. Table 5 shows the ratio of the total number of infections and the maximum number of daily infections compared to $R_0 = 3.0$ (purple curve in Fig 3), and the number of days to reach the maximum number of daily infections. Compared to the case of $R_0 = 3.0$, when $R_0$ is 1.5, the number of infected individuals dropped to 61.96%. Similarly, when $R_0 = 2.0$ and $R_0 = 2.5$, respectively, the number of infected individuals reduced to 84.74% and 94.91%. This result shows that the smaller the $R_0$, the lower the total number of infected individuals.

**Table 4. A summary of commonly used parameter values.**

| Parameters | Values | Remark and reference |
|:---:|:---:|:---:|
| $N$ | 5,000,000 | The population of Korea |
| $\Lambda$ | 788 | Daily average of the number of births in 2019 and 2020 [48] |
| $\nu$ | 5.7 | Mortality rate (persons per 1,000 population) [48] |
| $1/\sigma$ | 4.1 | The progression rate in Korea [46] |
| $1/\mu$ | 4 | The mean period from symptom onset to isolation in Korea [47] |
| $1/\gamma_1, 1/\gamma_2, 1/\gamma_3$ | 1/14, 1/23, 1/30 | Assumed by [49] |
| $\eta_1, \eta_2, \eta_3$ | 1/80, 1/12, 1/8 | Assumed |

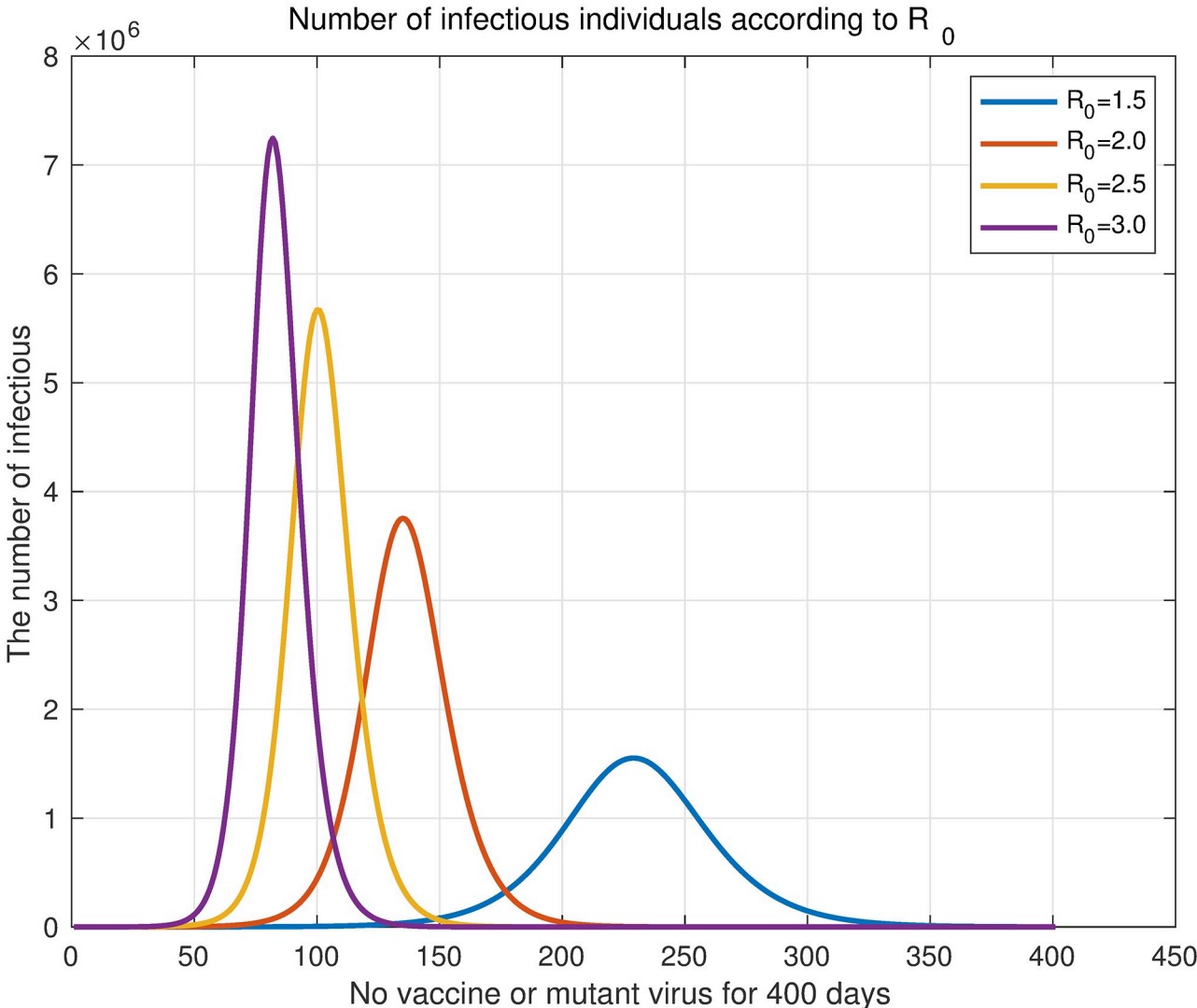

**Fig 3. Daily original virus infections according to $R_0$.** Four curves of changing $R_0$ from 1.5 to 3 in increments of 0.5. The purple curve shows the number of daily infections when $R_0 = 3.0$, and the yellow, red, and blue curves show when $R_0 = 2.5$, 2.0, and 1.5, respectively.

**Table 5. Infection rates by $R_0$.**

| $R_0$ | Ratio of total number of infections compared to $R_0 = 3.0$ | Ratio of maximum number of daily infections compared to $R_0 = 3.0$ | Number of days to reach maximum number of daily infections |
|---|---|---|---|
| 1.5 | 0.6196 | 0.2141 | after 229 days |
| 2.0 | 0.8472 | 0.5182 | after 135 days |
| 2.5 | 0.9491 | 0.7823 | after 100 days |
| 3.0 | 1.0000 | 1.0000 | after 82 days |

Table 5 notes the ratio of total number of infections and the maximum number of daily infections compared to $R_0 = 3.0$, and number of days to reach maximum number of daily infections.

An early study of COVID-19 by Kucharski et al. reported that the median of the basic reproduction number in Wuhan, China, fell from 2.35 (95% CI 1.15–4.77) to 1.05 (0.41–2.39) a week after travel restrictions were implemented [50]. This confirms that social distancing measures, e.g., travel restrictions, lower the basic reproduction number. Specifically, in terms of the effects of the decrease in the basic reproduction number, if $R_0 = 1.5$, then the maximum number of infected individuals decreased to 21.41% compared with the case of $R_0 = 3.0$. It decreased to 51.82% for $R_0 = 2.0$ and 78.23% for $R_0 = 2.5$. In addition, if the $R_0$ can be halved from 3 to 1.5, it was found that the number of days to reach the maximum number of infected individuals will be delayed to more than doubled. These results suggest that lowering the basic reproduction number by implementing non-pharmaceutical interventions in the early stage of infectious diseases plays a very important role in preventing the spread of infectious diseases.

## Case 2: Original virus with vaccination

To examine the relationship between vaccinations and the number of infected individuals, two simulations were conducted. For these, we set the incoming or outgoing parameters in $C^m$ associated with the mutant virus to 0. As reported in the previous subsection, we set the values of parameters are shown in Table 4. First, we changed the number of vaccinations per day to analyze the effect on the number of infected individuals for fixed $R_0$. Assuming that the number of vaccinated individuals is constant every day, the dosage per vaccination is selected to allow the entire population to complete the first vaccination within the target period. If we assume $k$ is target period to allow the entire population to complete the first vaccination, then $N/k$ is the number of daily vaccination. For notational convenience, we denote the parameter $1/k$ as $VD$ (Vaccination; Daily). In our experiment, the target periods were 100 days, 200 days, and 300 days with the second vaccination being 1/4 that of the primary vaccination. Booster shots were excluded. This determined the $\alpha_1(t)$, $\alpha_2(t)$, and $\alpha_3(t)$ parameters of the model. We set the probabilities of not forming antibodies $\delta_1$ and $\delta_2$ to 4.7% and 0.2%, respectively, referring to the paper [39] by Shrotri, M., et al. We also assumed that vaccination starts 100 days after the outbreak of an infectious disease.

In Fig 4, the blue curve represents the unvaccinated state while the red, yellow, and purple curves represent the number of vaccine doses as $N/300$, $N/200$, and $N/100$, respectively. Fig 4 shows the result obtained by changing $R_0$ from 1.5 to 2.3 at 0.1 intervals. As $R_0$ increases, the three curves (the red, yellow, and purple curves in each panel) representing the different numbers of doses per day gradually converge to the unvaccinated state.

The case of $R_0 = 2.0$ is examined for more details, which are shown in Fig 5. For $VD = 1/300$, $VD = 1/200$, and $VD = 1/100$, the number of infected individuals decreased to 77.37%,

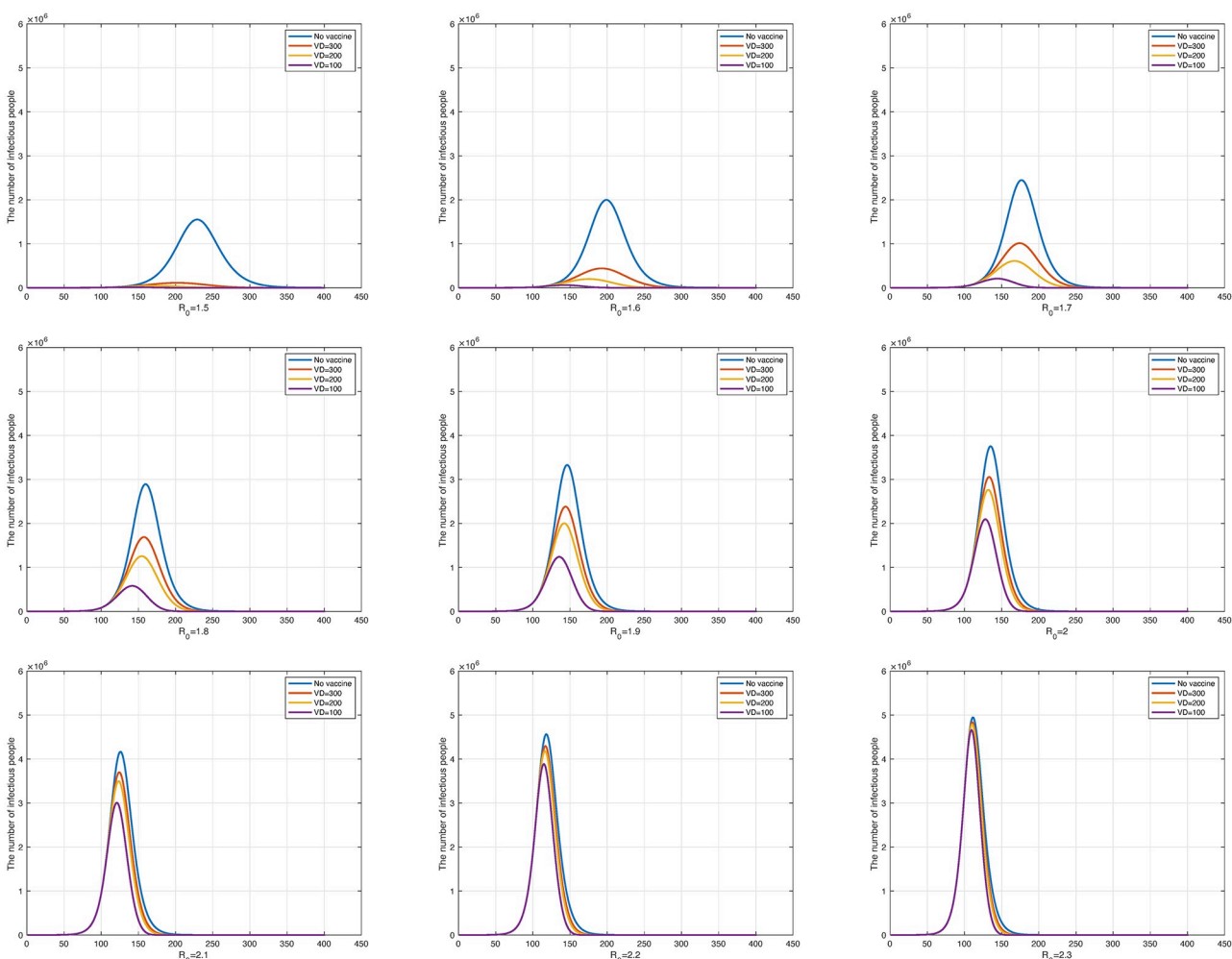

**Fig 4. The number of daily infections according to the change in $R_0$ from 1.5 to 2.3.** The upper left graph is $R_0 = 1.5$ and the lower right graph is $R_0 = 2.3$.

67.94%, and 46.66%, respectively, compared with the unvaccinated state (blue curve). In addition, as vaccinations increased, the number of infected individuals quickly reached the maximum daily infection rate, and the end of the infectious disease advanced more rapidly. This is because vaccination reduces the number of susceptible individuals who can become infected. The reduction in susceptible individuals accelerates the peak of infection and the onset of decline. Consequently, we can confirm through experiments that increasing the dose of vaccine can hasten the end of the disease.

Another parameter that can show the relationship between vaccination and the number of infected individuals is the time of the date when vaccination begins. By fixing the vaccine dose at $VD = 1/300$, we set the starting dates of vaccination to 100 days, 75 days, and 50 days after the outbreak. The purpose of this simulation was to examine the effects on the number of infected individuals when starting vaccination early. The total population and secondary vaccine dose continue under the same conditions as in the above experiments. Fig 6 shows the result obtained by changing $R_0$ from 1.5 to 2.3. The blue curves in the graphs represent the unvaccinated state, and the red, yellow, and purple curves the case with $VS = 100$, $VS = 75$, and

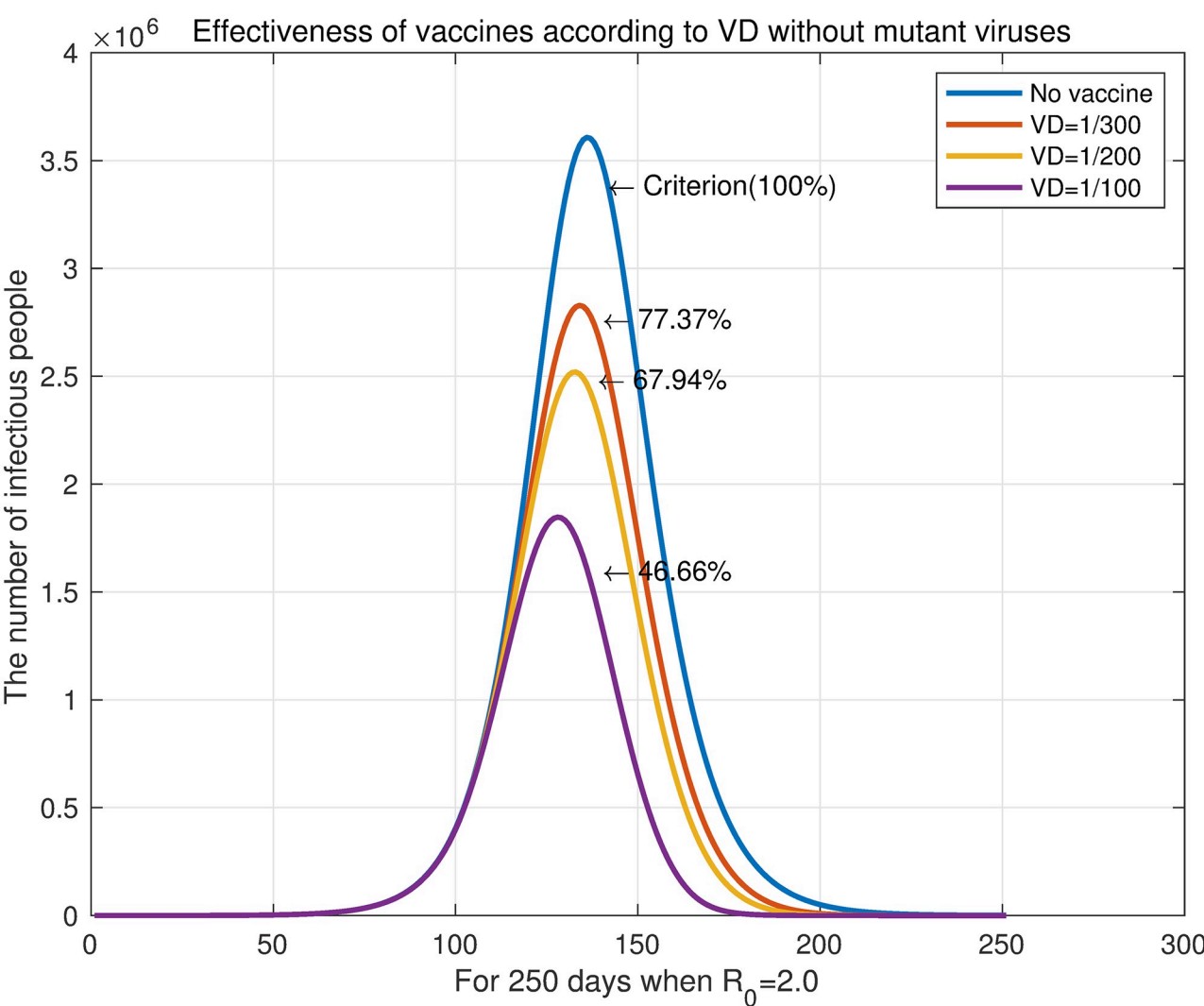

**Fig 5. The number of daily infections with $R_0 = 2.0$.**

$VS = 50$, respectively, where $VS$ indicates the date when vaccination begins after the outbreak. The results in Fig 6 indicate that the earlier the population starts being vaccinated, the lower the total number of infected individuals. In addition, as $R_0$ increased, we also observe convergence to the unvaccinated state just as in the experiment when the daily vaccinations was controlled.

We reviewed the case of $R_0 = 2.0$ in Fig 7. For $VS = 300$, $VS = 200$, and $VS = 100$, the number of infected individuals decreased to 77.37%, 61.42%, and 41.85%, respectively, compared with the unvaccinated state (blue curve). The graph shows that the earlier the vaccination, the later the maximum daily infections is reached. This contrasts with the effect of the number of daily vaccination on the maximum daily infection. The reason for this is the difference in the rate at which the susceptible individuals decrease according to the fixed basic reproduction number. Increasing the daily vaccination is effective in reducing the number of susceptible individuals. However, when the daily vaccination is kept constant, advancing the time of vaccination works differently from increasing the the number of daily vaccination. It only slows the

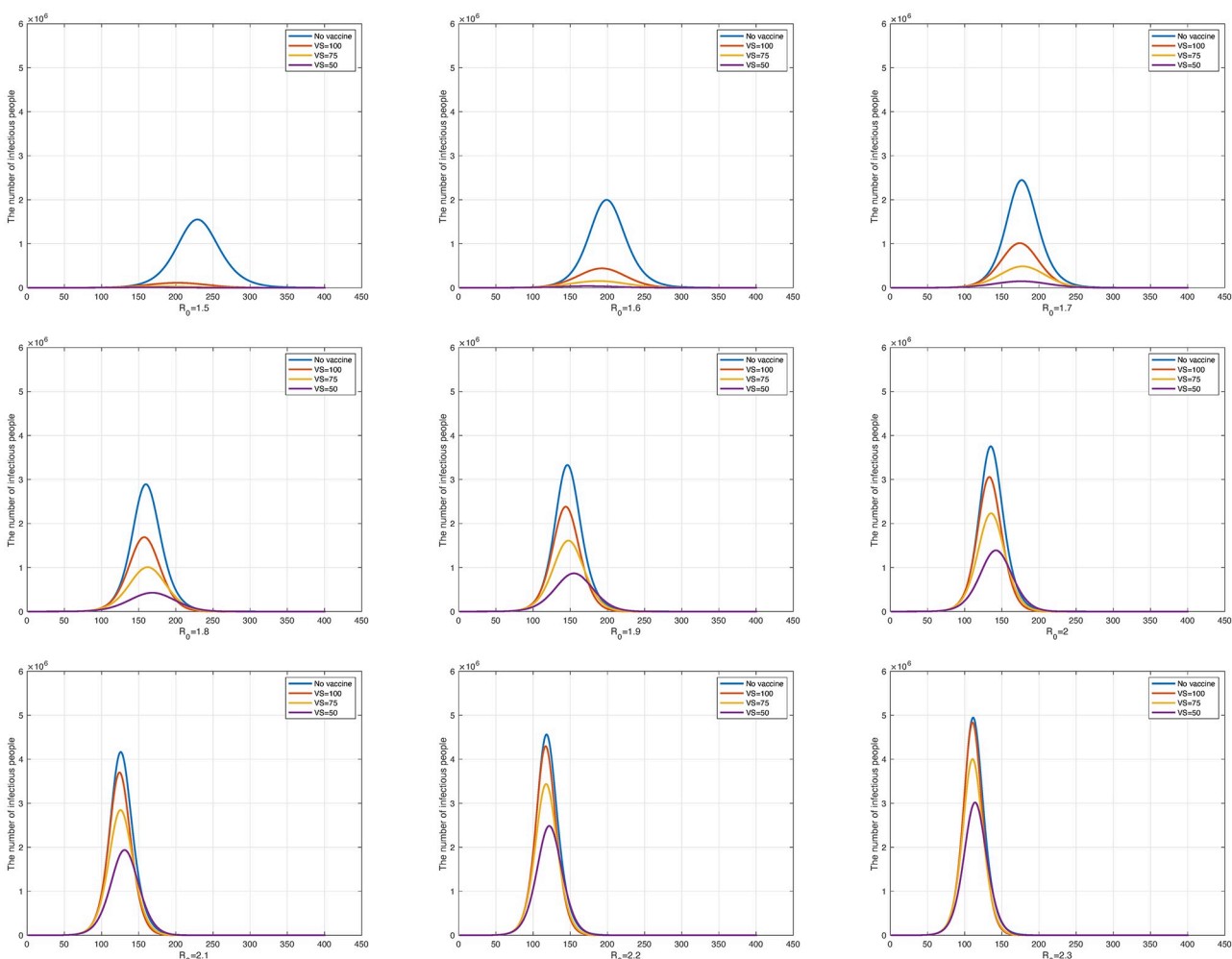

**Fig 6. The number of daily infections according to the change in $R_0$ from 1.5 to 2.3.** The upper left panel is $R_0 = 1.5$ and the lower right panel is $R_0 = 2.3$.

spread of the epidemic by starting to reduce the number of susceptible groups before the number of infectious groups grows. This is the same principle as slowing down the spread of infectious diseases through non-pharmaceutical interventions, and these changes in the graph are termed "flattening the curve". In our experiment, delaying the spread of an infectious disease also delayed the end of the epidemic.

We then experimented with the date vaccination is initiated and the number of daily vaccination. The parameters except for the number of daily vaccination ($VD$ parameter) and the starting date of vaccination ($VS$ parameter) follow the values set previously. In this experiment, we set the total number of infected individuals when $VD = 1/300$ and $VS = 50$ as the benchmark, and calculated the $VD$ that maintained the benchmark for the total number of infected individuals while changing the starting date of vaccination. This allowed us to simulate the amount of vaccine needed if vaccination is delayed.

In Table 6, we summarize $1/VD$ according to $VS$ when $R_0$ is 1.5 and 1.8, respectively, and Fig 8 is a graph showing it. The points marked with the blue marker "o" in Fig 8 are the measured $1/VD$ values for each $VS$, and the red curves are graphs obtained by fitting the given

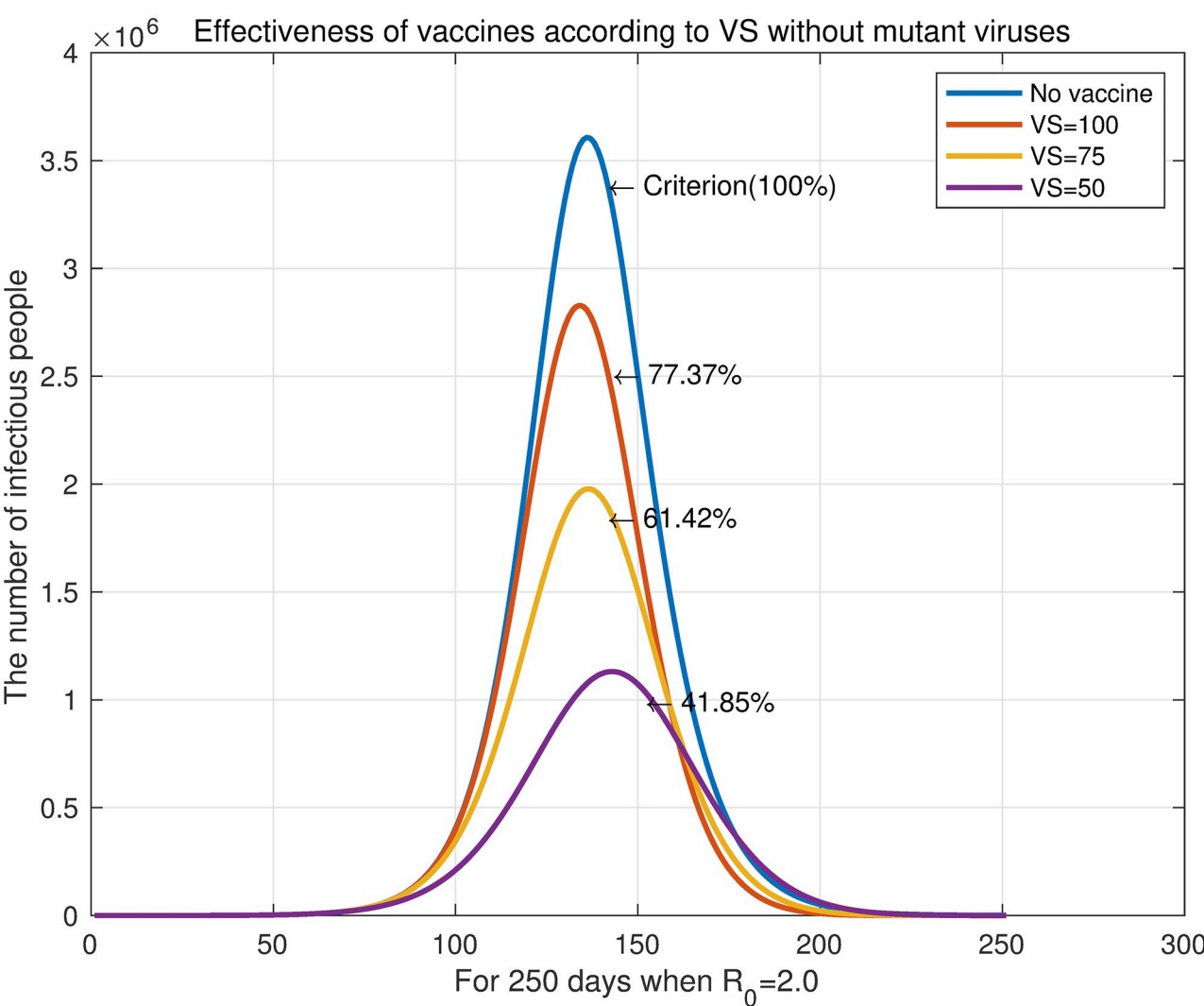

**Fig 7. The number of daily infections with $R_0 = 2.0$.**

points with a quadratic function. The RMSE between the measured values and the quadratic functions were 2.7964 and 2.0797 when $R_0$ was 1.5 and 1.8, respectively. Fig 8 shows that the period to allow the entire population to complete the first vaccination required to keep the total number of infections constant was decreased quadratically as *VS* increased. In other words, as the vaccination start time is delayed, the number of required daily vaccinations increases quadratically.

To express this numerically, if *VS* increases from 50 to 60 when $R_0 = 1.5$, the daily vaccination volume must increase by 30,000 doses, and if *VS* increases from 50 to 100, the daily vaccination volume must increase by 380,000 doses. Since $VD = 1/300$ when $VS = 50$, then the daily vaccinations would be 50, 000, 000/300 = 166, 667. This rate is similar to the average daily rate (144,516 doses) for 294 days from March 1 to December 31, 2021 in Korea. In the third quarter of 2021, the average daily vaccination volume in Korea was the highest. If *VS* increases from 50 to 100, 546,667 vaccinations per day would be required, which is more than double the daily rate of 254,033 vaccinations per day in Korea in the third quarter of 2021.

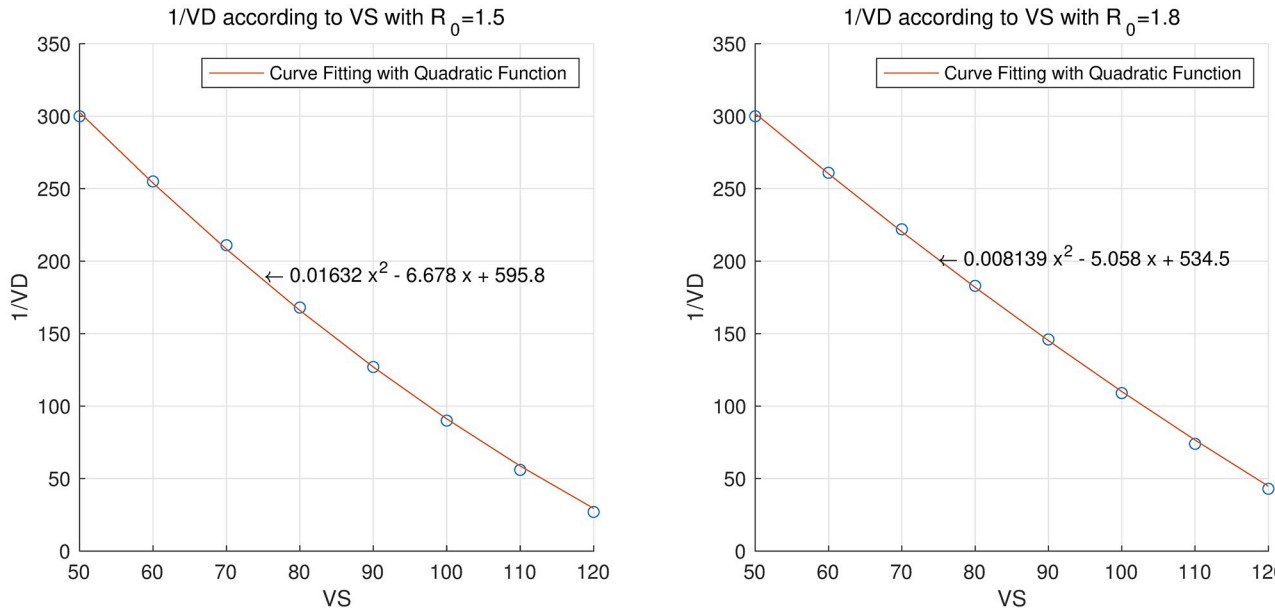

**Fig 8. 1/VD for each value of VS when $R_0$ = 1.5 or 1.8 and quadratic equations that approximates these points.** The left panel is when $R_0$ = 1.5 and the right panel is when $R_0$ = 1.8.

## Case 3: Original virus with mutant virus and vaccinations

In this simulation, the mutant virus was added after the start of vaccination. Similar to the simulations of Case 2, we experimented on the number of daily vaccination and the starting date of vaccination. First, we simulated the effect of the number of daily vaccination on the number of infected individuals when there is a mutant virus. We assume that vaccination starts 100 days ($VS$ = 100) and the mutant virus appears 200 days after the outbreak. In this experiment, booster shots were taken into consideration and it was assumed that inoculation was as much as $\alpha_1(t)/8$. The antibody formation rate of the vaccine was set identically to Case 2. Also, we refer to the data presented by the KDCA press release [41] for the antibody formation rate of the vaccine against the mutant virus. The values used were summarized in the Table 7. $R_0$ and $\tau$ were fixed at 1.3 and 1.8, respectively, and the number of vaccination was changed to $VD$ = 1/280, $VD$ = 1/300, and $VD$ = 1/320. The result is shown in Fig 9. This result confirms that the total number of infected individuals decreased as the number of daily vaccination increased.

Then, we fixed the number of daily vaccination to $VD$ = 1/300 and changed the time parameter of initiating vaccination. We fixed $\tau$ = 1.8 and changed to $VS$ = 75, $VS$ = 100, and $VS$ = 125. The results are shown in Fig 10, which confirm that the earlier vaccination begins,

**Table 6. 1/VD for each value of VS when $R_0$ = 1.5 or 1.8.**

| $R_0$ = 1.5<br>$VS$ = 50, $VD$ = 1/300 (criterion) | $VS$ | 60 | 70 | 80 | 90 | 100 | 110 | 120 |
|---|---|---|---|---|---|---|---|---|
| | $1/VD$ | 255 | 211 | 168 | 127 | 90 | 56 | 27 |
| $R_0$ = 1.8<br>$VS$ = 50, $VD$ = 1/300 (criterion) | $VS$ | 60 | 70 | 80 | 90 | 100 | 110 | 120 |
| | $1/VD$ | 261 | 222 | 183 | 146 | 109 | 74 | 43 |

In Table 6, we summarize $1/VD$ (the period to allow the entire population to complete the first vaccination) according to $VS$ when $R_0$ is 1.5 and 1.8, respectively.

**Table 7. The antibody formation rate.**

| Parameters | Values | Reference |
|---|---|---|
| $\delta_1, \delta_2, \delta_3$ | 4.7%, 0.2%, 0.1% | [39] |
| $\delta_1^m, \delta_2^m, \delta_3^m$ | 40%, 30%, 20% | [41] |

the lower the total number of infected individuals. However, the results regarding the end of the epidemic are different from Fig 7 in the previous subsection. Fig 7 shows that the earlier vaccination starts, the later it ends. In Fig 10, the sooner the vaccination starts, the faster the epidemic ends. This is due to the target period for total population to get their primary vaccination. According to Fig 10, a pandemic with a mutant virus lasted for approximately 600 days. Conversely, the simulated pandemic in Fig 7 lasted for 250 days. In this simulation,

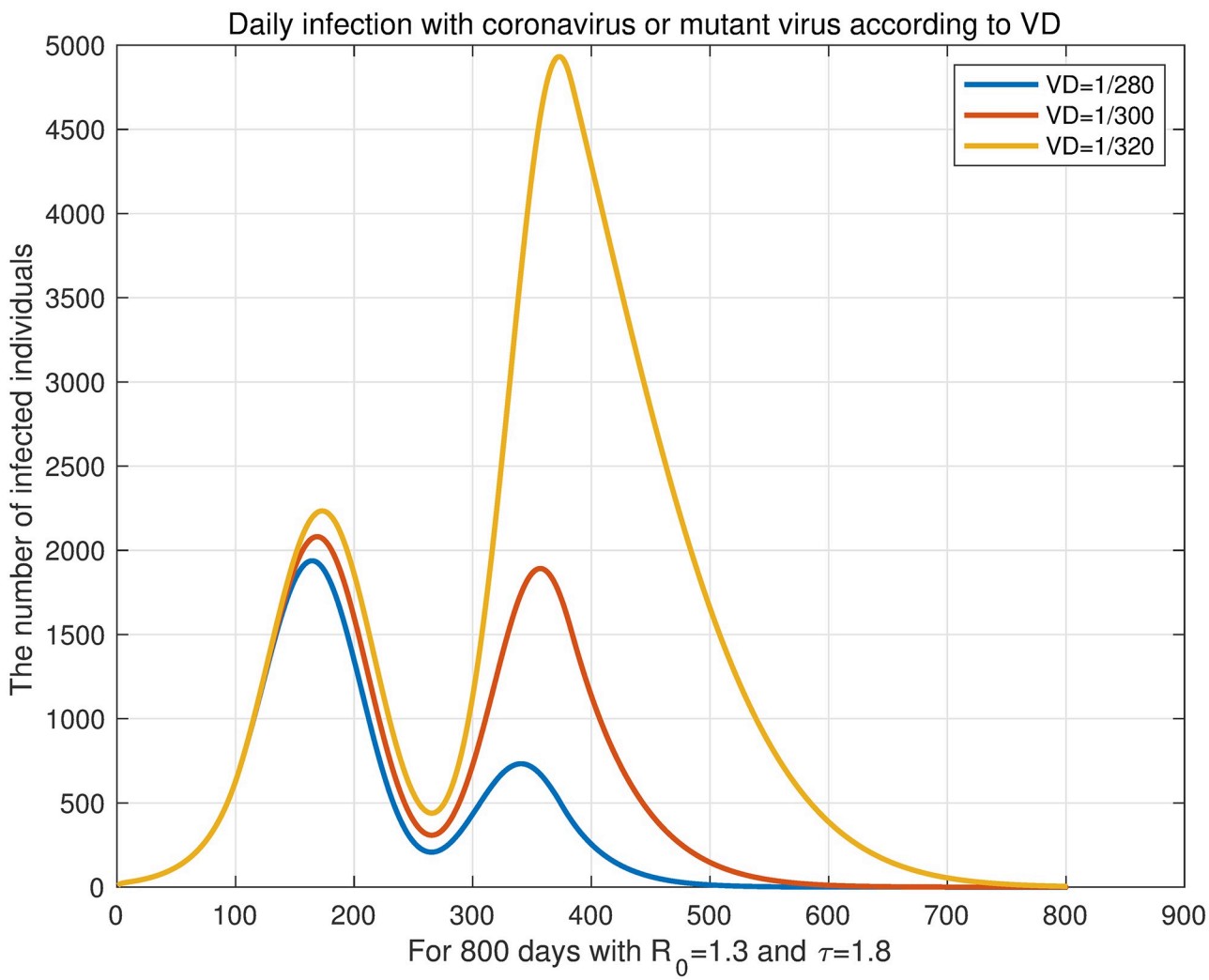

**Fig 9. Daily infection with coronavirus or mutant virus according to *VD*.** We assume that vaccination starts 100 days (*VS* = 100) and the mutant virus appears 200 days after the outbreak. $R_0$ and $\tau$ are fixed at 1.3 and 1.8, respectively, and the number of daily vaccination is changed to *VD* = 1/280, *VD* = 1/300, *VD* = 1/320.

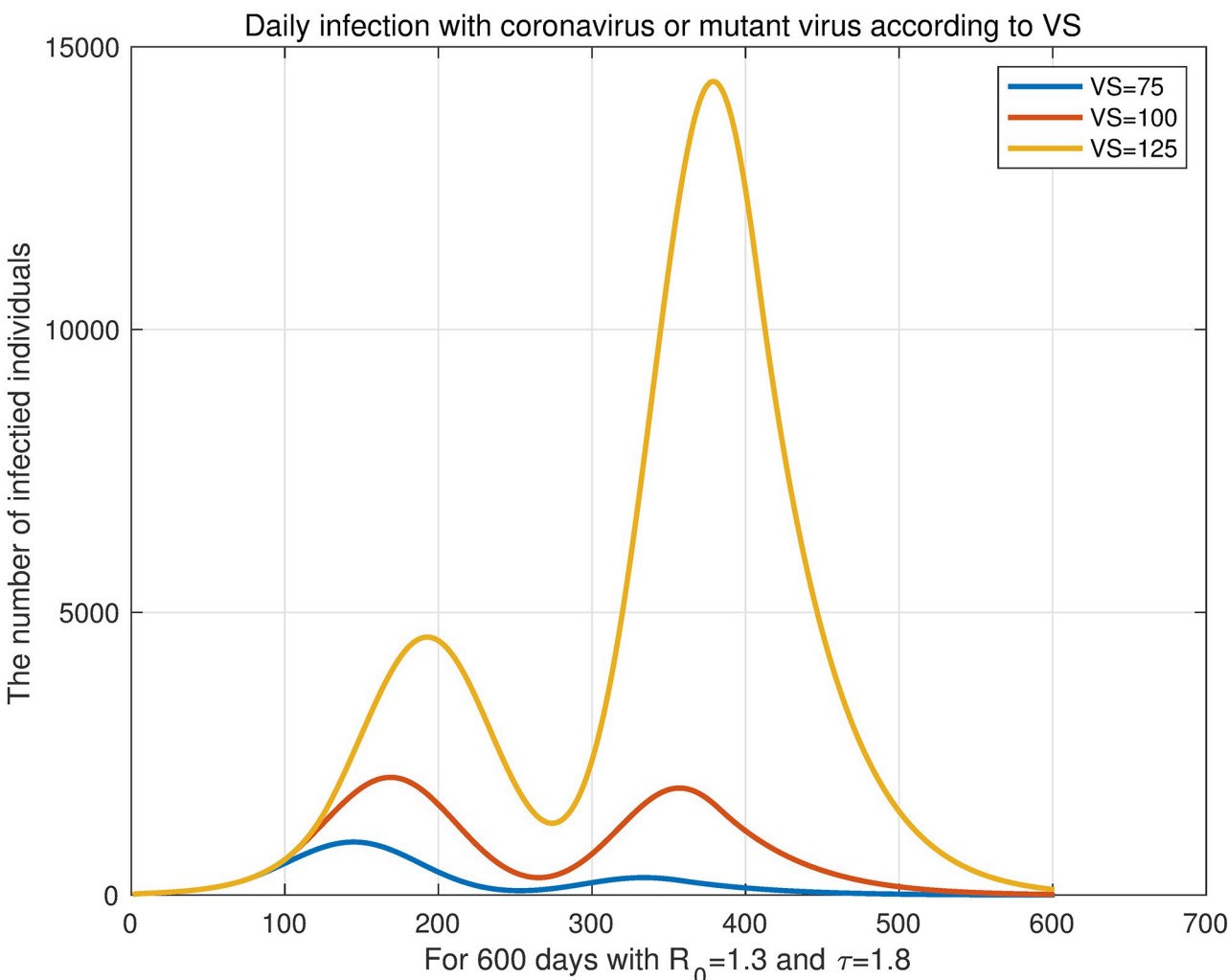

**Fig 10. Daily infection with coronavirus or mutant virus according to VS.** We fix the vaccine dose as $VD = 1/300$ and change the time parameter of starting the vaccination. For fixed $R_0 = 1.3$ and $\tau = 1.8$, change to $VS = 75$, $VS = 100$, $VS = 125$.

where an epidemic lasted for 600 days, the earlier the vaccination started, the sooner the epidemic ended because we assumed that the primary vaccination would end on day 300 ($VD = 1/300$).

Next, we examined the change in the total number of infected individuals according to the mutant virus's rate of transmission. We fixed the initial value of $R_0$ to 1.4 and changed $\tau$ from 1.3 to 1.8. The number of daily vaccination was set to $VD = 1/300$, and it was assumed that vaccination started 100 days after the outbreak. In addition, it was assumed that the mutant virus appeared from the 200th day. Here, since we fixed the transmission rate of the original virus and the susceptible groups gradually decreased with vaccination, following Eq (8), $R_t$ decreased over time. Similarly, under the same assumption, following Eq (10), $R_t^m$, the effective reproduction number of the mutant virus also reduced. Fig 11 shows the daily number of infected individuals and the change in effective reproduction numbers. Also, the effective reproduction number of the mutant virus is shown from the 200th day which is when the mutant virus appeared.

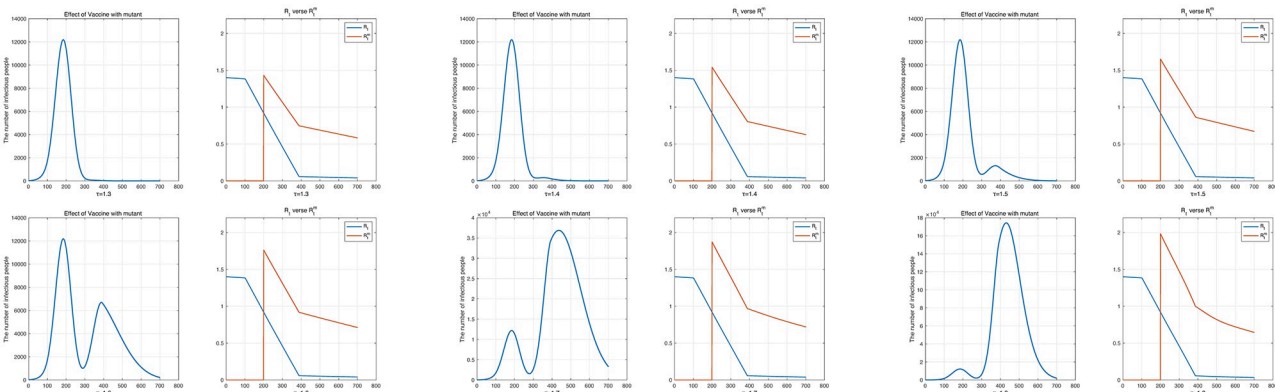

**Fig 11. The number of daily infected individuals with $R_0$ = 1.4 and $\tau$ ranging from 1.3 to 1.8 and the effective reproduction numbers of the original virus and the mutant virus according to each result.** The upper left panel shows $\tau$ = 1.3 and the 8th panel shows $\tau$ = 1.8.

Table 8 shows the ratios of the original and mutant virus infections to the total population according to $\tau$ and the effective reproduction numbers of each virus when the mutant virus appeared (Day 200 of Fig 11]). Then, since only the parameter $\tau$ related to the mutant virus was changed, the ratio of the number of infected individuals with the original virus to the total population and $R_t$ were fixed. Table 8, shows that as $\tau$ increased, the ratio of the number of mutant virus infections to the total population increased exponentially. However, when $\tau$ = 1.3, the largest effective reproduction number of the mutant virus was less than $R_0$ = 1.4 of the original virus, as shown in Fig 11. In this case, the mutant virus could not cause an outbreak. In addition, when $\tau$ was 1.7 or higher, $R_t^m$ was more than double that of $R_t$, and simulations confirmed that it became a turning point that greatly increased the number of mutant virus infections.

## Model fitting and prediction

We fitted Korea's data on COVID-19 infections to the *SVEIHRM* model and discussed the applications of the model in this section. Data from the KDCA [51] was used in the model fitting process. The original virus data were from February 16, 2020, to December 31, 2021, and up to this point Delta was the main mutant variant. Therefore, in this section, mutant virus refers to the Delta variant. Assuming that the transmission rate of the mutant virus is

**Table 8. The ratios of original virus and mutant virus infections to the total population.**

| $\tau$ | The ratio of the number of individuals infected with the original virus as the total population | The ratio of the number of individuals infected with the mutant virus as the total population | $R_t$ | $R_t^m$ |
|---|---|---|---|---|
| 1.3 | 2.55% | 0.02% | 0.9193 | 1.4314 |
| 1.4 | 2.55% | 0.07% | 0.9193 | 1.5415 |
| 1.5 | 2.55% | 0.37% | 0.9193 | 1.6516 |
| 1.6 | 2.55% | 2.46% | 0.9193 | 1.7617 |
| 1.7 | 2.55% | 17.00% | 0.9193 | 1.8718 |
| 1.8 | 2.55% | 60.41% | 0.9193 | 1.9819 |

Table 8 connects the ratios of the original and mutant virus infections to the total population according to $\tau$ and the effective reproduction number of each virus when the mutant virus appeared (Day 200 of Fig 11). We set the basic reproduction number to $R_0$ = 1.4 and changed $\tau$ from 1.3 to 1.8.

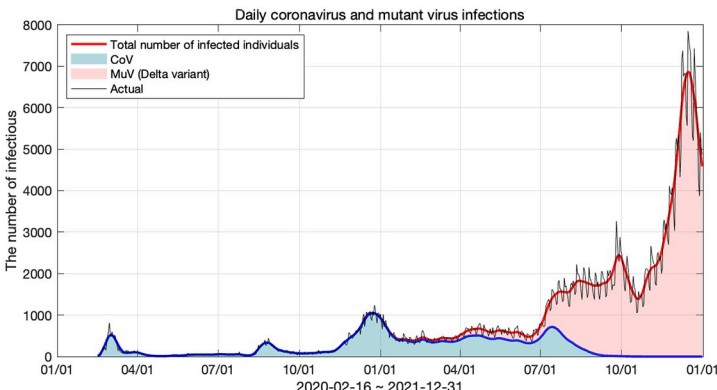
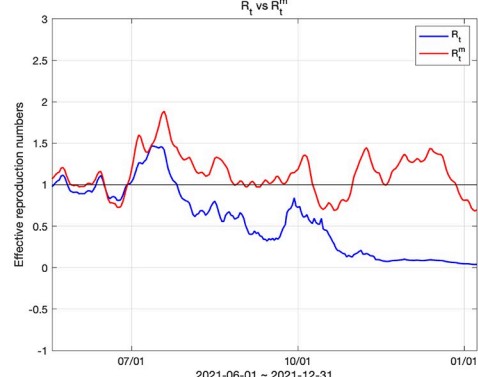

**Fig 12. Daily original virus and the Delta variant infections and effective reproduction numbers for the original virus and the Delta variant.** The left panel shows the number of daily infections and the right panel shows the effective reproduction number of the original virus and Delta variant.

maintained, we predicted the number of infected individuals with the original virus and the Delta variant. As application example using transmission rate obtained from model fitting, we prepared three scenarios according to the vaccination rate. Additionally, we simulated our model by applying it to an Omicron variant.

**Model fitting for Korea coronavirus data and reproduction numbers.** For model fitting, we undertook two processes. First, to fit the curve to the number of COVID-19-infected individuals, the daily data of infected individuals were smoothed using the cubic smoothing spline method. After that, the daily $\beta$ and $\tau$ values were calculated using the `least square fitting` functions of `MATLAB`. We calculated the $R_t$ and $R_t^m$ defined above using the daily $\beta$ and $\tau$ values obtained through model fitting and analyzed the current status of COVID-19 in Korea.

The graph in the left panel of Fig 12 shows the results of fitting our model to Korean coronavirus data from February 16, 2020 to December 31, 2021. $R_t^m$ and $R_t$ that were obtained by the model fitting are respectively indicated by the red and blue curves in the right panel. The reproduction numbers were calculated for the period from June 2021 to December 2021, and before that $R_t^m$ was 0 because no case of mutant virus infection had yet been reported. We measured the fitting error using the $L^2$ relative error norm and its value was 0.1364.

On the left graph in Fig 12, the blue area represents the original virus and the red area represents the mutant virus. According to the data of infected individuals from the KDCA [41], the number of individuals infected with the original virus decreased after August 2021, and then approached 0. Accordingly, $R_t$ of the original virus remained below 1 since August 2021. However, the total number of individuals infected with the original or mutant viruses showed an increasing trend, since the number of individuals infected with the mutant virus increased. The maximum $R_t^m$ for July, when the mutant virus started increasing significantly was is 1.89, which is close to 2. After that, $R_t^m$ decreased in October due to social distancing in Korea being strengthened but increased to 1.43 again as soon as social distancing was weakened. At this time, the number of infected individuals increased to nearly 8,000. After the second week of December, the total number of COVID-19 cases began to decline again. There are two main reasons for this. One is the strengthening of social distancing and the other is the decline of the virus due to a decrease in the number of susceptible individuals. This analysis is discussed in the next subsection on prediction using our models.

**Table 9. Weekly average effective reproduction numbers for December 2021.**

| Period | Average value of $R_t^m$ |
|---|---|
| 12/04 ∼ 12/10 | 1.3820 |
| 12/11 ∼ 12/17 | 1.2917 |
| 12/18 ∼ 12/24 | 0.9910 |
| 12/25 ∼ 12/31 | 0.7783 |
| Average $R_t^m$ | 1.1108 |
| Average value of $\tau$ | 3.3736 |

Table 9 notes the weekly average $R_t^m$ and average value of $\tau$ for December 2021.

**Prediction of the COVID-19 (the original virus and the Delta variant) infections in Korea.** We estimated the number of Delta variant infections using the model fitting results from the previous subsection to calculate the weekly average effective reproduction numbers for December 2021. The results are shown in Table 9 below.

We used the average $R_t^m$ of December 2021 as the Delta variant's reproduction number proceeding into the future. The average value of $R_t^m$ for 4 weeks was 1.1108, and based on experience, the range of change in $R_t^m$ in the next month was set to ±10%. Therefore, we set the value of $R_t^m$ from 0.9997 to 1.2219 as the reproduction number of the virus to be processed in the future. Using these, the prediction results for up to the next 100 days are shown in Fig 13.

Fig 13 tells us that the number of infected individuals decreased even though the reproduction number was 1 or higher. The reason is that because more than 80% of the population had already been vaccinated, the number of susceptible individuals was small. According to the

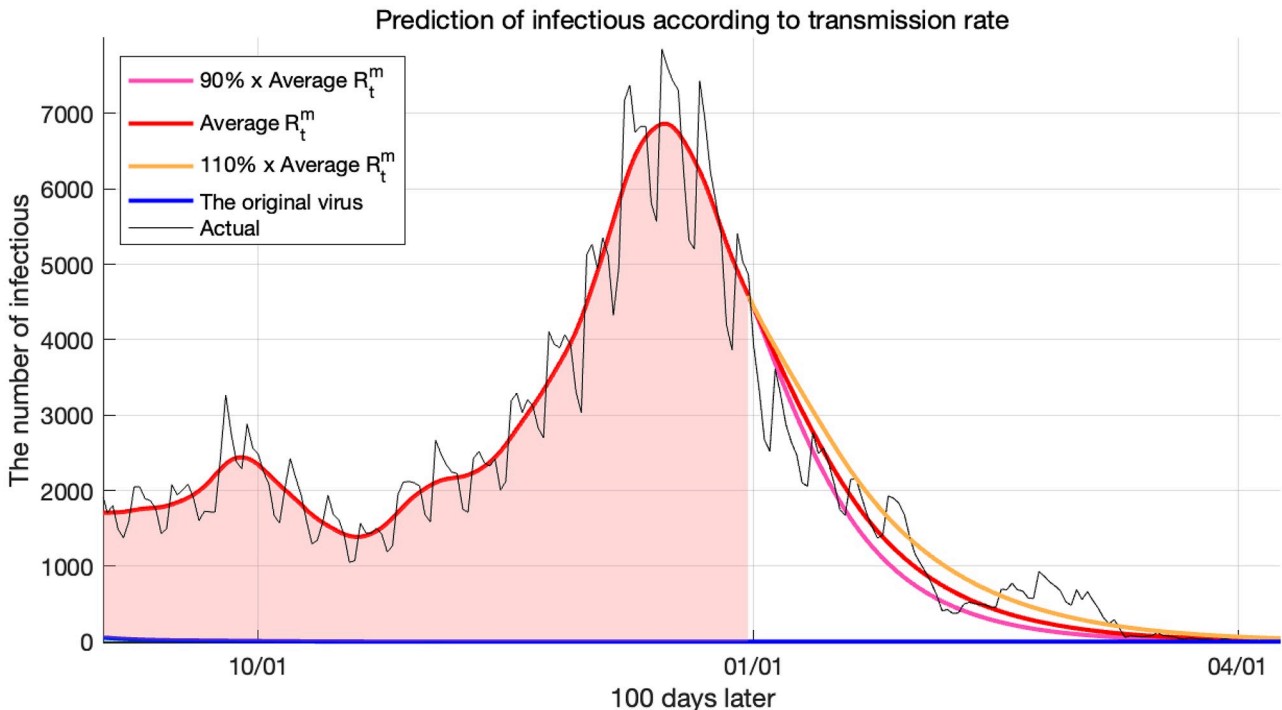

**Fig 13. Prediction of infections for the original virus and the Delta variant according to reproduction numbers.**

derived $R_t^m$ (10), even if the transmission rate $\beta^m$ value is large, if the number of individuals belonging to the compartment $S$ is small, the effective reproduction number decreases quickly. Therefore, the reason for the decrease in virus infections, from December 2021, was that although the strengthening of social distancing was also effective, the vaccination rate was higher than this.

According to our simulation results, it was expected that the Delta variant would end between February and March 2022. The black curve in Fig 13 represents the number of individuals infected with the actual Delta variant. We see that it is consistent with our predictions for the Delta variant, hence confirming that our model predicted the trend of the Delta variant well. The fitting errors measured by the $L^2$ relative error norm were 0.2521 for the average $R_t^m$, and 0.2888 and 0.2642 for 90% and 110% of the average $R_t^m$, respectively.

## Sensitivity analysis of changes in vaccination rate

Our model is capable of adjusting various parameters. We adjusted the inoculation rate of the vaccine and analyzed the number of infected individuals accordingly. The vaccination rate was controlled by adjusting the $\alpha_1(t)$ parameter related to the movement from the susceptible group to the first vaccination and the $\alpha_2(t)$ parameter, which is the ratio of the movement from the first vaccination group to the second vaccination group. Similarly, $\alpha_3(t)$ was also changed. We prepared three scenarios. First, the actual vaccination rate in Korea was used as a reference. In the other two simulations, $\alpha_1(t)$, $\alpha_2(t)$, and $\alpha_3(t)$ were multiplied by 0.9 to represent a 10% reduction in vaccinations and multiplied by 1.1 to represent a 10% increase in vaccinations. The results are shown in Fig 14.

The graph in Fig 14 shows the period from June to December 2021, when the mutant virus started to increase. The red curve in Fig 14 is the curve of the number of infected individuals according to the actual inoculation rate in Korea. The blue and yellow curves respectively represent the 10% reduction and 10% increase in vaccinations. Table 10 shows the ratio of the total number of infected individuals to the actual total number of infected individuals in each scenario. Otherwise expressed, if the vaccination rate was increased by 10% by December 2021, the number of infected individuals would decrease by 35.22%, and conversely, if the vaccination rate was reduced by 10%, it was expected that the number of infected individuals would increase by 82.82%.

## Model fitting for the Delta variant and the Omicron variant

The spread of the Omicron variant can be simulated by modifying our model. Since the original virus has already been terminated, if the Delta variant is placed in the original virus compartment and the Omicron variant in the mutant virus compartment, we obtain a modified model. Accordingly, $\delta_k$, which is the parameter of the original model, is replaced with $\delta_k^d$, and $\delta_k^m$ is replaced with $\delta_k^o$. The antibody formation rate for the Omicron variant is the value reported by the KDCA [41].

In Korea, the Omicron variant began being reported in the last week of November 2021. We fitted the Omicron variant and the Delta variant to the modified model from the last week of November 2021 to the first week of February 2022. When fitting the model, the initial value of the population belonging to each compartment could not be known because the modified model was not simulated from the beginning of the epidemic. First, the vaccination compartments were estimated using the cumulative values of the primary and secondary vaccinations and booster shots. Since this model was intended to simulate the Omicron variant, the populations of $V_k$ were set as $\delta_k^o$ times the cumulative dose of the primary and secondary vaccinations and booster shots for $k = 1, 2, 3$, respectively. In addition, the infectious compartments and

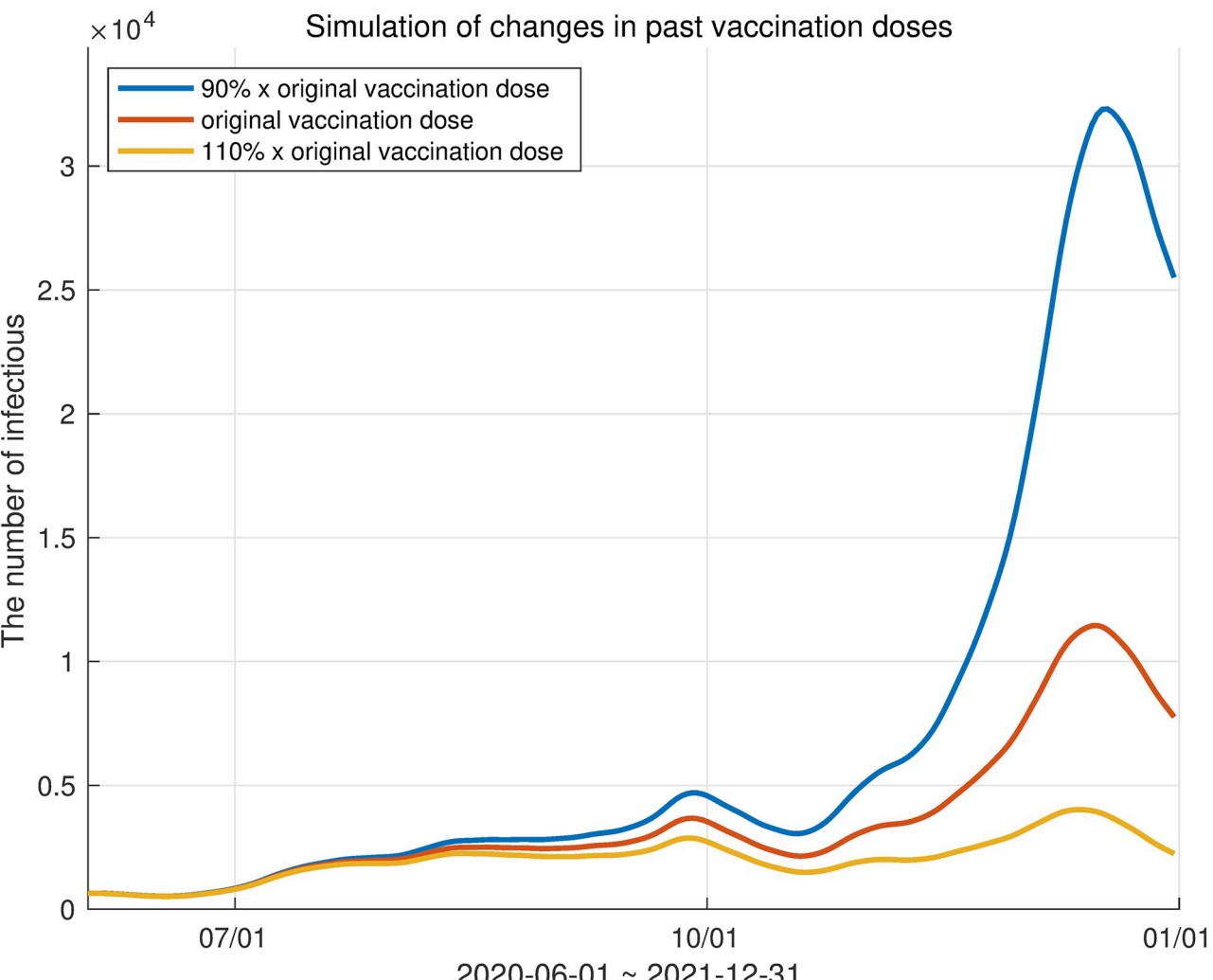

**Fig 14. Three scenarios depending on vaccine dose.** The red curve represents the number of infected individuals according to the actual inoculation rate in Korea. The blue and yellow curves represent the 10% reduction and 10% increase in vaccination, respectively.

exposed compartments were set by the number of infected individuals in the last week of November 2021. However, it was difficult to estimate the number of populations in each compartment from the data we had. Therefore, the remaining populations, except for the allocated population from the total number of populations, were placed in the susceptible compartment, and the remaining compartments were initialized to 0. The result of the model fitting is shown in Fig 15.

**Table 10. The ratio of the total number of infected individuals in each scenario to actual infections.**

|  | ratio |
| --- | --- |
| 10% reduction in vaccine dose | 1.8282 |
| 10% increase in vaccine dose | 0.6478 |

Table 10 notes the ratio of the total number of infected individuals in each scenario (blue and yellow curves in Fig 14) to the actual total number of infected individuals (red curve in Fig 14).

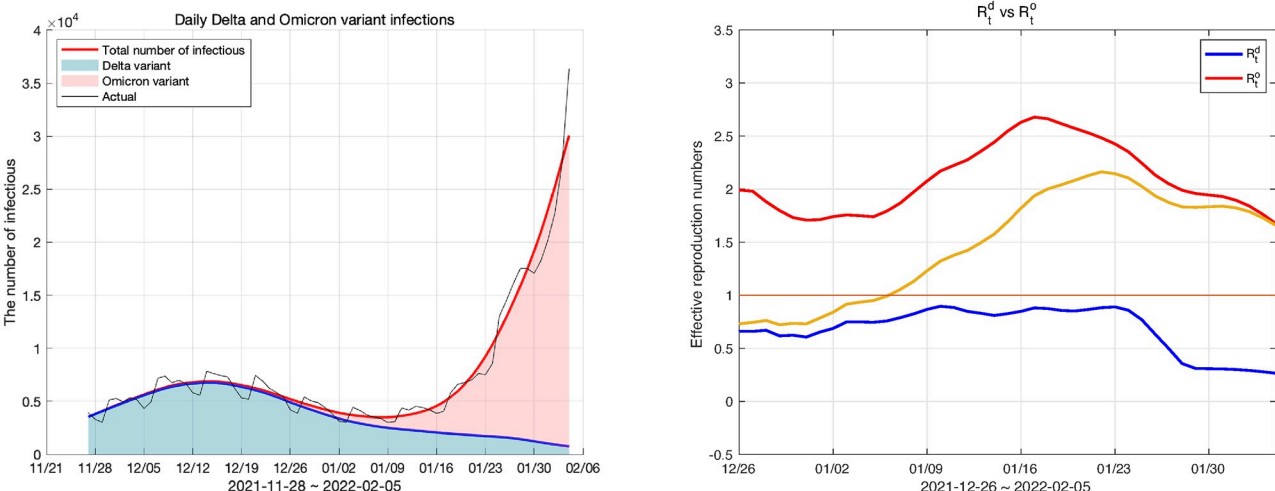

**Fig 15. Daily Delta and Omicron variant infections and effective reproduction numbers for the Delta and the Omicron variants.** The left panel shows the number of daily infections and the right panel shows the effective reproduction number of the Delta and Omicron variants.

The fitting results from the last week of November 2021 to the first week of February 2022 are shown in Fig 15. However, in model fitting, the initial value of the reproduction number is unreliable because the setting of the initial value is not accurate. Therefore, we show the reproduction numbers from one month after the population began to be properly placed in each compartment. In the left panel of Fig 15 the red area represents the daily Delta variant infections, and the green area represents the Omicron variant's daily infections. Also, in the right panel in Fig 15, the red curve indicates the effective reproduction number of the Omicron variant and the blue curve indicates that of the Delta variant. The yellow curve in this figure is the average of the two reproduction numbers weighted by their shares of total infections. The fitting error calculated by the $L^2$ relative error norm was 0.1275.

In the left panel of Fig 15, we can see that the number of individuals infected with the Delta variant gradually decreased while the number infected with the Omicron variant increased exponentially. The Omicron variant became dominant from the second week of January 2022, and at this time, the effective reproduction number of the Omicron variant approached 2.5. The Omicron variant became the dominant variant in one and a half months. We can explain this observation using the effective reproduction number. From the beginning of the outbreak of the Omicron variant to December 2021, its share of infections was insignificant, but the effective reproduction number at this time was close to 2. This was a larger number than the reproduction number in July when, as shown in Fig 12, the Delta variant started to surge. Additionally, while the Delta variant maintained a high effective reproduction number for a short period, the Omicron variant maintained a reproduction number of 1.5 or higher for more than a month. This result means that the infectivity of the Omicron variant is greater than that of the Delta variant. Therefore, at the end of December 2021, although the number of infected individuals of the Omicron variant was low, the reproduction number was higher than any other variant of COVID-19 because it is highly transmissive. Consequently, the number of Omicron variant infections increased rapidly.

## Discussion

We presented a compartment model that can simulate pandemics with mutant viruses and multiple vaccinations. We experimented with three cases: 1) the original virus only without

vaccination, 2) the original virus with vaccination, and 3) the original virus with a mutant and vaccination. In Case 1, there was neither a mutant virus nor a vaccine, so we focus on the control of $R_0$. The results of this experiment represented the effect of non-pharmaceutical interventions such as social distancing in the early stage of the COVID-19 outbreak. We found in this experiment that non-pharmaceutical interventions lowered the number of confirmed cases and slowed the spread of transmission. A case where $R_0$ was actually reduced by the implemented policy was introduced in [50] and strategies for non-pharmaceutical interventions was introduced in [8]. Cases 2 and 3 focused on the effect of the number of daily vaccinations and the timing of the start of vaccination on the number of infected individuals. The total number of infected individuals decreased as the vaccination was increased or the vaccination starting time was earlier. However, these two variables caused different results when it came to the transmission of infectious diseases. The higher the number of daily vaccinations, the faster the daily number of infected individuals reached its maximum, leading to the early ending of the epidemic. The earlier vaccination started, the slower the maximum daily number of infected individuals was reached, delaying the end of the epidemic. It may sound somewhat paradoxical that the earlier vaccination starts, the more it delays the end of the epidemic. However, this is the same principle as "the lower the basic reproduction number, the slower the spread of the infectious disease and the slower the end." Therefore, early vaccination flattens the COVID-19 curve as shown in Fig 7 and helps us to avoid overwhelming the medical facilities in early stages of the pandemic. It also has the advantage of buying time to obtain a sufficient volume of vaccines, since the supply of vaccines has fallen far short of demand and an insufficient number of vaccinations will delay the end of COVID-19. Securing as many vaccines as possible and increasing the vaccination rate are recommended to minimize the damage of the pandemic. Even with vaccination, non-pharmaceutical interventions are still important. Figs 4 and 6 show that with increasing $R_0$, we observe convergence to the unvaccinated state even with vaccination. As a detailed study related to this, there is a study [9].

We also compared the transmission rates of the original virus and the mutant virus as shown in Fig 11 and Table 8. An interesting result of this experiment was that a new peak was produced when the effective reproduction number of the mutant virus was more than double that of the original virus. This could help us to allocate medical and administrative resources to two different viruses in the presence of mutant viruses. Of course, this result is sensitive to the characteristics of mutant viruses and by how much vaccination is delayed. Also, since we used only Korean data for this experiment, we should be careful with generalizations. We leave this for future research agenda.

Given real data, the other use of this model is to find the reproduction number of the epidemic. For this, we founded the basic and the effective reproduction number of the *SVEIHRM* model. We fitted the model to Korean COVID-19 data and calculated the reproduction numbers of each virus. We used these effective reproduction numbers to account for the change in the number of confirmed cases in Korea from June to December 2021. This was consistent with the results of strengthening and weakening social distancing. Using the fitting results, our model predicted the number of future infections. By correlating the predictions with the effective reproduction number of mutant virus, we analyzed that the decline in the Delta variant beginning at the end of December 2021 was attributable to vaccination rates.

We simulated three scenarios by changing the vaccination rate. In this simulation, we predicted the number of infected individuals depending on the scenarios. As a result, we showed that a 10% change in vaccination rates can make a huge difference, reducing the number of confirmed cases by 35.22% or increasing it by 82.82%. These results may help inform people who are hesitant to get vaccinated.

We acknowledge some limitations of the present study. Our model ruled out reinfection between the two strain because the rate of reinfection was low at the time of the Delta variant in Korea [52]. We conducted simulations for the Omicron variant, which was newly reported toward the end of November 2021, by changing the parameters of the model and confirmed that the Omicron variant has a very large effective reproduction number, as shown in Fig 15. However, reinfection rates of the Omicron variant have been reported to be high [52] and our model was not suitable to provide a fitting and prediction for this variant in the long term. Also, we did not consider antibodies that disappear over time. This is very important for mutant viruses with high reinfection rates that may emerge in the future. We leave the model improvement considering reinfections for future research.

An economic cost and benefit analysis of mitigating policies such as social distancing and vaccination can be conducted using the results of various simulations in this study. Specifically, we can examine whether the economic benefits of social distancing (or vaccination) exceed or fall short of its costs. Since the Korean government has changed the intensity of social distancing several times in the course of fighting the spread of COVID-19, it can be a natural experiment to estimate the benefits and costs of the various intensity of social distancing.

Economic benefits of social distancing (or vaccination) mainly come from the decreased number of infected individuals and the number of deaths due to the mitigating policy. A recent economic study on COVID-19 (Thunström, L. et al., 2020 [53]) performs a cost and benefit analysis of social distancing measures, considering the value of reduced mortality risk as a benefit and a decrease in GDP as a cost of social distancing measures. The study extrapolates the estimates of the influence of social distancing taken to combat the spread of the 1918 Spanish flu for the examination of COVID-19. Due to globalization and technological development since then, the current economic environment is significantly different from what it was 100 years ago, so it does not seem to be appropriate to apply the estimates at that time to the present study.

However, in the case of COVID-19, it is difficult to estimate the decrease in GDP due to social distancing since governments and central banks worldwide alleviated the decline in GDP by expanding money supply and fiscal expenditure. It would be possible to estimate the decline in GDP due to the pandemic more accurately when the process of economic downturn come to an end due to a series of inflations. We leave it as an agenda for future research.

## Supporting information

**S1 File. Codes used to get raw data using OpenAPI.** These codes are programmed in ipynb format.
(ZIP)

**S1 Text. An explanation to the construction of the model.**
(PDF)

**S2 Text. Proof of proposition 1.**
(PDF)

## Author Contributions

**Conceptualization:** Young Rock Kim, Yong-Jae Choi, Youngho Min.

**Data curation:** Youngho Min.

**Formal analysis:** Youngho Min.

**Investigation:** Youngho Min.

**Methodology:** Young Rock Kim, Yong-Jae Choi, Youngho Min.

**Project administration:** Young Rock Kim.

**Software:** Youngho Min.

**Supervision:** Young Rock Kim.

**Validation:** Young Rock Kim, Yong-Jae Choi.

**Visualization:** Youngho Min.

**Writing – original draft:** Youngho Min.

**Writing – review & editing:** Young Rock Kim, Yong-Jae Choi, Youngho Min.

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
