## [Decision Letter · Decision Letter 0]

21 Jul 2022

PONE-D-22-14438A model of COVID-19 pandemic with vaccines and mutant virusesPLOS ONE

Dear Dr. Min,

Thank you for submitting your manuscript to PLOS ONE. After careful consideration, we feel that it has merit but does not fully meet PLOS ONE’s publication criteria as it currently stands. Therefore, we invite you to submit a revised version of the manuscript that addresses the points raised during the review process.

We look forward to receiving your revised manuscript.

Kind regards,

Martial L Ndeffo Mbah, Ph.D

Academic Editor

PLOS ONE

Journal Requirements:

2. In your Introduction, please ensure that all statements are clearly supported by reference to the existing body of peer-reviewed literature, or are otherwise supported by the data in the manuscript, in order not to appear as subjective. Please remove any statements that cannot be supported in this way. Examples include, but are not limited to, "The Korean government failed to procure sufficient amount of vaccines" (line 24-5).

Please also note that PLOS ONE has specific guidelines on code sharing for submissions in which author-generated code underpins the findings in the manuscript. In these cases, all author-generated code must be made available without restrictions upon publication of the work. Please review our guidelines at https://journals.plos.org/plosone/s/materials-and-software-sharing#loc-sharing-code and ensure that your code is shared in a way that follows best practice and facilitates reproducibility and reuse.

Additional Editor Comments:

Reviewers have raised several concerns regarding both methodology and results of the manuscript. Thoroughly addressing these comments will greatly improve the quality of the manuscript and it suitability for publication. Though novelty is not a requirement for publication in PLoS ONE, we agree with Reviewer 4 that if there is any novelty in the manuscript, it should be clearly highlighted and discuss how the current analysis extend or complement similar published analyses.

Reviewers' comments:

Reviewer's Responses to Questions

**Comments to the Author**

1. Is the manuscript technically sound, and do the data support the conclusions?

Reviewer #1: Yes

Reviewer #2: Yes

Reviewer #3: Partly

Reviewer #4: Yes

2. Has the statistical analysis been performed appropriately and rigorously? 

Reviewer #1: N/A

Reviewer #2: Yes

Reviewer #3: Yes

Reviewer #4: Yes

3. Have the authors made all data underlying the findings in their manuscript fully available?

Reviewer #1: Yes

Reviewer #2: Yes

Reviewer #3: Yes

Reviewer #4: Yes

4. Is the manuscript presented in an intelligible fashion and written in standard English?

Reviewer #1: Yes

Reviewer #2: Yes

Reviewer #3: No

Reviewer #4: Yes

5. Review Comments to the Author

Reviewer #1: The authors proposed an extended SIR with mutation and multi-vaccine model.

They claim to fit the model for the case of Koria for COVID-19.

In summary, the idea is good and implementation is well done. However, the text should be improved.

I hope my comments would help the authors to improve their work.

Major:

1. The text is hard to read due to the grametical and lingustic issues currently found in the paper. The authors suggested to use a native English speaker to fix these issues in the text.

2. The review pargaraph in the Introduction section is limited. The authors need to disucss recent works about multi-strain and multi-mutation models and more models that combine SIR and vaccination.

In particualr, this journal and others have several recent such publications:

A. Marquioni and de Aguiar, Modeling neutral viral mutations in the spread of SARS-CoV-2 epidemics (2021) - Plos one.

B. Lazebnik and Bunimovich-Mendrazitsky, Generic Approach For Mathematical Model of Multi-Strain Pandemics, (2022) - Plos one.

C. Gordo et al., Genetic Diversity in the {SIR} Model of Pathogen Evolution (2009) - Plos One.

D. Arruda et al., Modelling and optimal control of multi strain epidemics, with application to COVID-19 (2021) - Plos one.

E. Lazebnik and Blumrosen, Advanced Multi-Mutation with Intervention Policies Pandemic Model (2022) - IEEE Acsses.

F. Khyar and Allali, Global dynamics of a multi-strain SEIR epidemic model with general incidence rates: application to COVID-19 pandemic (2020) - Nonlinear Dynamics.

3. The novelty of the work is not clear from the Abstract or the Introduction sections.

4. "It assigns a population to labeled compartments – for example, S, I, or R, 81 (Susceptible, Infectious, or Recovered)." - bad wording and cite! this is not your idea...

5. I do not see any point in giving so much attention to Eqs. 1 + 2, you can remove this sub-section...

6. An explanation to the construction of Eq. (3) should be provided (even in the appendix).

7. Proposition 1 - we saw similar results in multiple works in the past - seems redundant to me.

8. Line 281 - why? you should explain these parameter values.

9. The results can be presented better. A lot of the graphs repeating themselves and do not provide any additinal insights. The authors should present a "meta" analysis of these results to the reader.

10. I think the "Cost and benefit analysis of vaccines" is out of the scope of the paper and the authors tried to introduce another "cool" result. I reccomend to delete it.

11. The Discussion section just "reads" the results - it does not provide any insights to how policymakers can use this model and results.

12. The Discussion section lack an honest discussion of the research limitations.

13. A fitting error analysis is missing.

14. I would like to see a "novelty statment" comparing the current work with previous multi-strain extended SIR models. I think the authors would see that the current "novelty statement" is somewhat problematic.

Minor:

1. Intro, first statement - cite?

2. Intro, second statement - cite?

3. Intro, "Using the above model, we provide various simulations" - no model is provided yet at this point.

4. Intro, last paragraph is not clear to me. It seems you are all over the place and I was not able to follow your logical process.

5. "There are various methods for mathematical modeling of infectious diseases." - cite?

6. "compartment model is one of the most representative mathematical modeling techniques" - cite?

7. "To construct our model, we start with basic epidemic models which are the prototypical epidemic models." - what?

8. "As an epidemic occurs, the number of susceptible individuals decrease over time and eventually becomes zero." - not ture, please see https://doi.org/10.3390/sym13071120

9. "the number of secondary infections produced by one infected individual after the initial outbreak is called the effective reproduction number." - cite?

Positive:

1. Really likes tables 1 + 2.

2. The "An effective reproduction number" sub-section well presented.

3. The "Model fitting for Korea Coronavirus data and reproduction numbers" sub-section is fairly presented.

Reviewer #2: The authors propose an extension of the SEIR compartment model to describe the evolution of the COVID-19 epidemics. Many additional compartments are added to the model to include vaccinations (two doses plus booster shot), hospitalizations and mutant variants. After a careful analysis of main features of the model the authors fit the parameters to data from Korea, obtaining interesting results.

When I started reading the paper I thought it would be messy and hard to follow, given the large number of parameters needed to accommodate all compartments and the corresponding transition rates between them. However, the paper is really well written and organized. I like the analysis presented in the Results section and the way conclusions are drawn from the figures. I also enjoyed reading the section where the model parameters are fit to the Korean data and how the original+delta strains are replaced by delta+omicron. I believe the model contributes significantly to advance the modeling of COVID-19 and deserves publication in PLoS One.

I have only a few minor suggestions that the authors should consider:

1 - item 3 of the "SEIHRV -mutant model" is confusing. I suggest simplifying it to

The vaccine of the coronavirus (CoV) is effective for mutant viruses (MuV) and cross-reinfections are ruled out.

2 - the model assumes the all infected individuals are hospitalized. However, most people recover without need to go to a hospital. The authors could, therefore, mention that the lowest level of hospitalization could also be interpreted as "no-hospitalization" (and no cost).

3 - I don't understand the need for Laplace transforms to prove proposition 1. The matrix in eq.(12) can be obtained directly from the coefficients of E and I in eq.(8), since the system is linear.

4 - I find the notation VD for vaccination dose a bit confusing. Because VD is defined as the number of days it takes to vaccinate the entire population, the larger is VD the slower is the rate of vaccination. I understand this is convenient for the authors but it might be clearer to use 1/VD or maybe 1000/VD to have numbers of order 1. That would have a more direct interpretation, as large VD would mean large the rate of vaccination. But I leave it to the authors whether it is worth to change the notation.

5 - line 370 - "Now we now"

6 - I don't think figure 14 is needed. Maybe it could me moved to the supplemental material.

7 - Figure 16 is very important. It is impressive how a change in 10% in the rate of vaccination could make such a large difference and probably save so many lives. This should be shown to all anti-vaccination groups.

Reviewer #3: The authors proposed an extended SIR with mutation and a multi-vaccine model.

In general, the idea is good and the implementation is on top. I like their work.

This paper has some publishing points, but there are several issues why this paper is not ready for publication.

Main:

1. Extensive editing of English language and style required. The text is difficult to read due to grammatical and linguistic problems found in the article.

2. The abstract does not report any findings, only an approach. What is the novelty of the manuscript?

3. The biological introduction and the review of existing modelling is insufficient.

The authors should discuss recent work on multi-strain, multi-mutation models and other models combining SIR and vaccination.

There are several recent publications in PLOS one:

D. Arruda et al., Modeling and optimal control of multi-strain epidemics with application to COVID-19 (2021) - Plos. one.

A. Marchioni and de Aguiar, Modeling Neutral Viral Mutations in the Spread of SARS-CoV-2 Epidemics (2021) - Plos one.

B. Lazebnik and Bunimovich-Mendrazitsky, A general approach to the mathematical model of multi-strain pandemics, (2022) - Plos one.

C. Gordo et al., Genetic Diversity in the {SIR} Model of Pathogen Evolution (2009) - Plos One.

4. "It allocates the population to labeled compartments, such as S, I, or R, 81 (susceptible, infectious, or recovered)." - bad wording and quote! it's not your idea...

5. The equations 1+2 are known, not need speak about it…

6. Explanation of the equations (3) must be provided!.

8. N = 50, 000, 000, σ = 1/4.1 and µ = ¼ - you must explain these parameters!

9. No clear explanation of the graphs. Authors should provide analysis of these results.

10. The section: Vaccine Cost Benefit Analysis not relevant to the purpose of this article...

11. Chapter "Discussion" is poorly motivated in terms of biological interpretation. it does not provide any insight into how policymakers might use the model and the results.

12. What is about the limitations in the Discussion?

13. There is no analysis of fitting errors.

Minor: the statements in the introduction require confirmation from the literature:

1.the first statement - reference?

2. second statement - reference?

3. Introduction: "Using the model above, we provide various simulations" - what model ?

4. The last paragraph in Introduction is not clear...

5. "There are various methods of mathematical modeling of infectious diseases." - reference?

6. "The coupe model is one of the most representative methods of mathematical modeling" - reference?

7. “To build our model, we will start with basic epidemic models, which are the prototypical epidemic models.” – not clear?

8. “As an epidemic occurs, the number of susceptible people decreases over time and eventually becomes zero.” – not agree…

Reviewer #4: Reviewer’s Comment

Dr. Shyam Sumanta Das

July 19, 2022

Article: PONE-D-22-14438

A model of COVID-19 pandemic with vaccines and mutant viruses

The author has worked on a model of the COVID-19 pandemic with the pres-

ence of multiple mutant viruses and studied the effect of vaccination on diease

propagation in South Korea. They have developed a mathematical model based

on the SEIR type of compartment model in which they have added the various

compartments related to multiple vaccinations and mutant virus of COVID-19.

The author has calculated the effective reproduction numbers and equilibrium

points of the original virus, Delta, and Omricon strains of the SARS-CoV-2

virus. Further, the author has carried out numerical simulations for the differ-

ent cases and fit the model with the observed data from the confirmed cases

from South Korea. The author has did a good work. I will suggest the following

modifications in the manuscript.

1. I will suggest the author to rewrite the introduction . In the introduc-

tion , the author should cite some more literatures related to modeling

of COVID-19 disease transmission with the presence of multiple mutant

viruses and the effect of vaccination. Also, the author should include some

works related to multistrain dynamics of various strains of SARS-CoV-2

virus.

2. In the method part, it is not necessary to explain about SIR and SEIR

model. The author can directly start explaining the proposed SEIHRV

model which is based on SEIR framework. They can cite the original

reference of SEIR model while writing the method.

3. The author should explain little bit about the breakthrough infection in

the model.

4. In Figure 1., the author should slightly modify the schematic diagram

for SEIHRV model. There is an extra I m compartment near the vaccine

compartments (V 1 ,V 2 and V 3 ).

5. As the COVID-19 pandemic is endemic now in the world, I think the

author should include the demographic information in model such as birth

rate and death rate.

6. In Figure. 11, the author should mentioned the values of τ each of the

sub-figures from 1.1 to 1.8.

6. PLOS authors have the option to publish the peer review history of their article (what does this mean?). If published, this will include your full peer review and any attached files.

Reviewer #1: **Yes: **Teddy Lazebnik

Reviewer #2: No

Reviewer #3: No

Reviewer #4: No

---

## [Author Response · Author response to Decision Letter 0]

5 Sep 2022

First of all, we would like to thank the reviewers and editors for the positive feedback and helpful comments on corrections or corrections. 

The first reviewer and the third reviewer recommended us researches on the multi-strain model as a reference. This contributed greatly to the revision of our manuscript. Thanks to them, we had the opportunity to reexamine the statements of the manuscript in detail. The second reviewer considered the results of our manuscript as important. Finally a 4th reviewer and editor suggested rewriting our introduction, which allowed us to improve our manuscript.

Detailed responses to each reviewer's comments are in the attached pdf file "Response on Reviewers".

---

## [Decision Letter · Decision Letter 1]

26 Sep 2022

A model of COVID-19 pandemic with vaccines and mutant viruses

PONE-D-22-14438R1

Dear Dr. Min,

We’re pleased to inform you that your manuscript has been judged scientifically suitable for publication and will be formally accepted for publication once it meets all outstanding technical requirements.

Kind regards,

Martial L Ndeffo Mbah, Ph.D

Academic Editor

PLOS ONE

Additional Editor Comments (optional):

Please, fix the minor typos identified by the reviewer and make sure to provide the missing figures.

Reviewers' comments:

Reviewer's Responses to Questions

**Comments to the Author**

1. If the authors have adequately addressed your comments raised in a previous round of review and you feel that this manuscript is now acceptable for publication, you may indicate that here to bypass the “Comments to the Author” section, enter your conflict of interest statement in the “Confidential to Editor” section, and submit your "Accept" recommendation.

Reviewer #1: All comments have been addressed

Reviewer #2: (No Response)

Reviewer #3: (No Response)

Reviewer #4: All comments have been addressed

2. Is the manuscript technically sound, and do the data support the conclusions?

Reviewer #1: Yes

Reviewer #2: Yes

Reviewer #3: Yes

Reviewer #4: Yes

3. Has the statistical analysis been performed appropriately and rigorously? 

Reviewer #1: Yes

Reviewer #2: Yes

Reviewer #3: N/A

Reviewer #4: Yes

4. Have the authors made all data underlying the findings in their manuscript fully available?

Reviewer #1: Yes

Reviewer #2: Yes

Reviewer #3: Yes

Reviewer #4: Yes

5. Is the manuscript presented in an intelligible fashion and written in standard English?

Reviewer #1: Yes

Reviewer #2: Yes

Reviewer #3: Yes

Reviewer #4: Yes

6. Review Comments to the Author

Reviewer #1: The authors wonderfully addressed my concerns and produce a high quality mathematical work, I believe it can be published in PLOS ONE

Reviewer #2: The authors did a very good job in revising the paper, tackling all the questions and

suggestions raised by the reviewers. I think this is a greatly improved paper. I have only

minor suggestions that the authors should consider.

1 - The abstract could be still more explicit about the achievements of the work. I suggest an extra

sentence highlighting the effects of start date of vaccination, something like

"We also show that start vaccinations early is key to reduce the number of infected individuals. Delaying the start date requires increasing substantially the rate of vaccination to achieve similar target results."

2 - There are still some minor English corrections needed. For instance:

- Abstract - "In particular, our model consider" -> In particular, our model considers

- page 8: A basic reproduction number -> The basic reproduction number

- page 9: An effective reproduction number -< The effective reproduction number

3 - Figures 3, 4 and 6 were missing in this version. I assume they are the same as in the

previous version.

Reviewer #3: no

Reviewer #4: Now, I am fully satisfied with the authors work. They have clearly addressed all my questions. I must recommend the work should be publish.

7. PLOS authors have the option to publish the peer review history of their article (what does this mean?). If published, this will include your full peer review and any attached files.

Reviewer #1: **Yes: **Teddy Lazebnik

Reviewer #2: No

Reviewer #3: No

Reviewer #4: No

---

## [Editor Report · Acceptance letter]

13 Oct 2022

PONE-D-22-14438R1 

A model of COVID-19 pandemic with vaccines and mutant viruses  

Dear Dr. Min:

I'm pleased to inform you that your manuscript has been deemed suitable for publication in PLOS ONE. Congratulations! Your manuscript is now with our production department. 

Kind regards, 

on behalf of

Dr. Martial L Ndeffo Mbah 

Academic Editor

PLOS ONE